# Saliency models perform best for women's and young adults' fixations

Christoph Strauch [1,2✉], Alex J. Hoogerbrugge [1,2], Gregor Baer[1], Ignace T. C. Hooge[1], Tanja C. W. Nijboer[1], Sjoerd M. Stuit[1] & Stefan Van der Stigchel[1]

Saliency models seek to predict fixation locations in (human) gaze behaviour. These are typically created to generalize across a wide range of visual scenes but validated using only a few participants. Generalizations across individuals are generally implied. We tested this implied generalization across people, not images, with gaze data of 1600 participants. Using a single, feature-rich image, we found shortcomings in the prediction of fixations across this diverse sample. Models performed optimally for women and participants aged 18-29. Furthermore, model predictions differed in performance from earlier to later fixations. Our findings show that gaze behavior towards low-level visual input varies across participants and reflects dynamic underlying processes. We conclude that modeling and understanding gaze behavior will require an approach which incorporates differences in gaze behavior across participants and fixations; validates generalizability; and has a critical eye to potential biases in training- and testing data.

[1] Experimental Psychology, Helmholtz Institute, Utrecht University, Utrecht, The Netherlands. [2] These authors contributed equally: Christoph Strauch, Alex J. Hoogerbrugge. ✉email: c.strauch@uu.nl

The world provides us with rich potential visual input. The immense amount of available information, combined with the unevenly distributed photoreceptor cells of the retina and the limited processing capacity of the visual system, necessitates several steps of prioritization. The first, and perhaps most important, of these steps determines how gaze, and with it visual attention, is allocated across a given scene. The way the eyes are rotated affects which visual information falls on the eyes' highly or lowly resolving parts, and herein lies the foundation of how individuals see and perceive the world. Which aspects of a scene are most salient - and thus determine where observers will likely fixate - is therefore of crucial interest[1].

In order to computationally model visual saliency, topographic maps of image features such as local orientation, contrasts, spatial frequencies, or colors are integrated - in other words features that are of importance in early visual areas within the visual cortex. Initial applications of these models were seen in computer vision, namely the prioritization of highly informative locations to deal with limited processing capacity of computers[1]. It was not long, however, until saliency maps were translated back to vision: Which locations of a scene, based on image features, will most likely attract covert and overt shifts of attention and thus be fixated to optimally use the brain's processing power[2]. These models can be understood as spatial distribution maps that highlight salient over non-salient areas. In turn, such maps can be compared to actual gaze data from benchmarking data sets[3,4] and improvements can be made to iteratively adapt models to become ever closer to gaze data, thereby also improving understanding of how low-level visual input might drive this behavior.

While the initial and seminal models were constructed by considering visual features that are well known to be represented in the early visual areas of the brain[2], dozens if not hundreds of models have since been proposed, some following similar approaches, some semantically enhanced [e.g., ref. [5]], and some based on deep learning approaches instead [e.g., ref. [6]]. Researchers in turn benchmark their models on empirical data which usually contain fixations of a limited number of adult participants [e.g., refs. [7–9]] who view a very wide range of static images without instruction [see[4], for an overview]. For instance, the influential and vast CAT2000 dataset[10] contains free viewing data of 120 participants (80 women, 40 men) aged 18–27 years with each of 4000 images being viewed for 5 s by 24 participants. This approach guarantees generalization across (static) stimuli. Recent calls for more diverse samples in (psychological) research[11–13] - beyond the overrepresented participant pools of most research universities[12] - raise the question of how well such findings may generalize across individuals rather than images. Just as much as demographic biases in training data might bias models, it may be asked whether saliency drives eye movements uniformly over time; that is, whether models are as predictive for

the first as for intermediate or later fixations. Both questions can only be studied with massive samples, as fixation maps of single fixations, such as all second fixations, are otherwise too scarce to allow for inferences. With the emergence of massive online studies, such generalizability issues are increasingly addressed for other questions in psychology. Large samples, however, remain scarce in eye tracking research. Whether the field of saliency modeling is also affected by said sampling biases as one of the key challenges of present-day psychology[11] is therefore still unclear.

Here, we tested the generalizability of model predictions across people and across fixations, not stimuli, and set out to uncover possible biases in model predictions with regard to both gender, age, and across fixations. To this end, we evaluated performance of 21 saliency models, selected upon availability and their huge influence on the field to infer conclusions on saliency mapping as a general discipline. For each saliency model, performance was assessed relative to a spatial distribution map of fixation locations of $n = 2607$ participants, including children, on a single image (see Fig. 1). As further baselines, we employed (1) a central bias, based on the assumption that participants fixate the center more than the periphery[14]; (2) a meaning map[15], constructed of relative meaningfulness ratings for small patches of the stimulus; and (3) a single observer baseline, the averaged predictivity of each participant's fixation locations for fixations of all other participants. Furthermore, we evaluated model performance across fixations - in other words how well early, intermediate, and later viewing behavior could be modeled. For analyses that relate model performance to demographic details of participants, gaze behavior of $n = 1600$ participants from 6 to 59 years of age and highest credibility of logged demographics were used.

## Results

**Descriptive model performance.** Normalized scanpath saliency (NSS) scores were used to evaluate model performance, a measure that has generally favorable properties and correlates highly to other indicators of model performance[16,17]. The continuous spatial distribution map of fixations of all participants yielded a NSS of 0.709 relative to the discrete fixation locations of all participants, effectively establishing the upper bound for any model's performance. Surprisingly, model performance was higher for many models (e.g., SALICON, SalGAN DeepGazeII) than for the meaning map (NSS = 0.382), which has been described to outperform regular saliency models[15]. The central bias had a prediction very close to chance level (NSS = 0.014) and was outperformed by all but one of the evaluated saliency models; the single observer was outperformed by most models. Performance of individual models is given in Table 2 with absolute NSS and percentage deviations in model performance scaled between central bias and the overall fixation map.

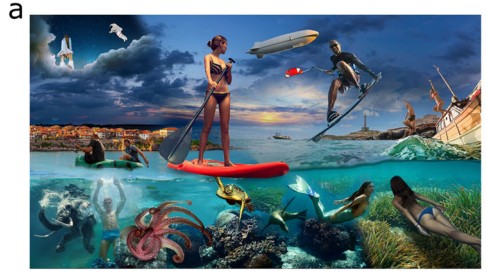
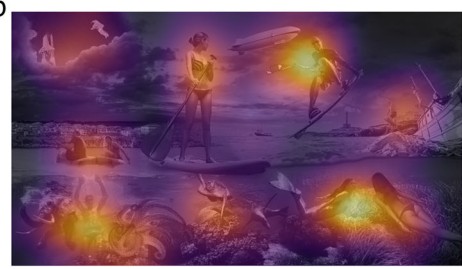

**Fig. 1 Stimulus and overlaid fixation map. a** Stimulus presented to assess free viewing data and **b** spatial distribution map of fixation locations overlaid with brighter colors indicating more fixations. Usable gaze data were obtained from $n = 2607$ participants using an eye tracker and monitor which were part of an installation at a science museum (see Supplementary Figure 1 for pictures of the setup). Collage made from licensed stock images from Shutterstock.

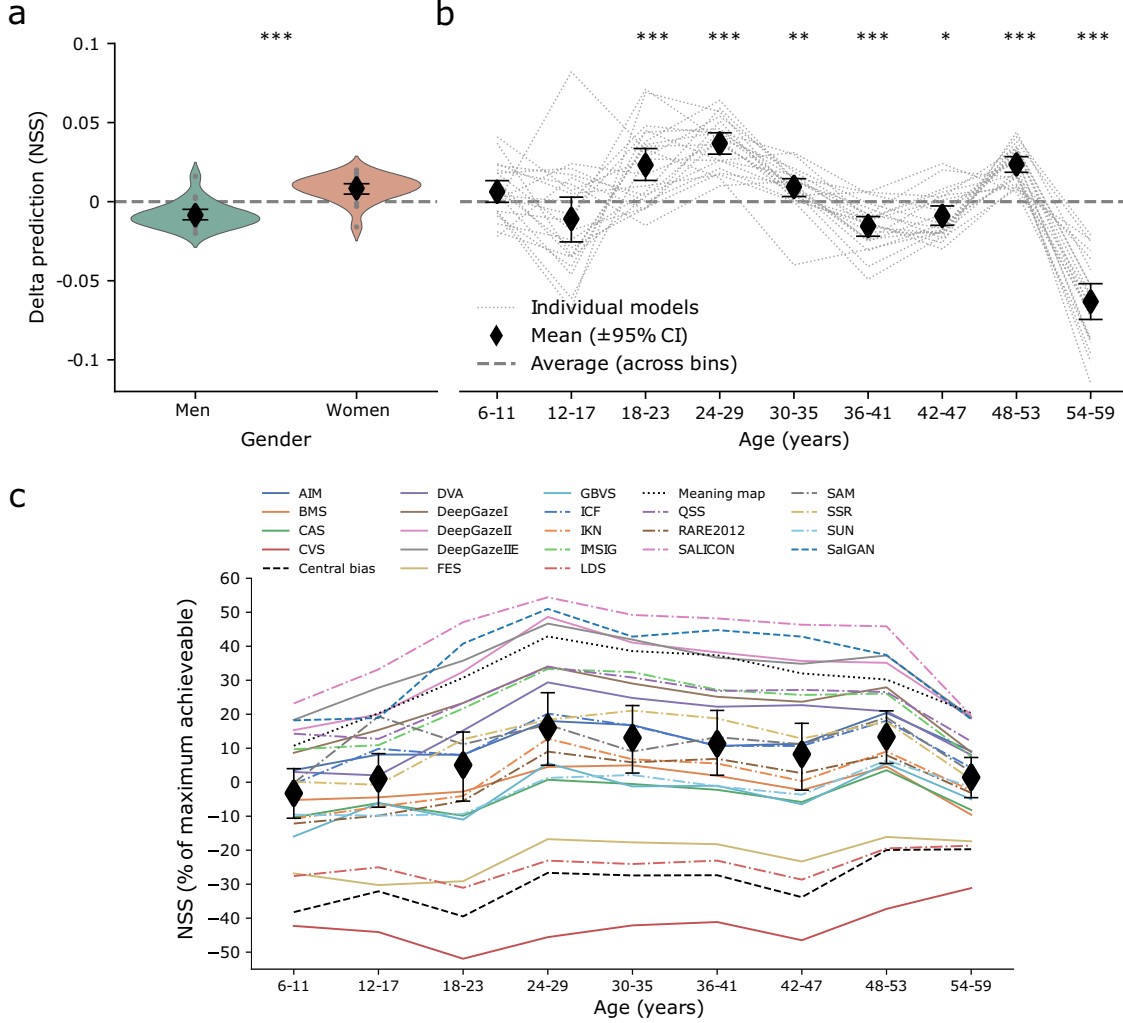

**Fig. 2 Model performance across demographic bins.** Positive and negative values indicate better- and worse than average performance, respectively. Individual data points represent NSS deviations from average model performance across age bins. **a** Deviations for men and women. **b** Deviations across age bins. Gray dots and lines depict NSS deviations for individual models. \*\*\*: statistically significant at $p < 0.001$, \*\* at $p < 0.01$, \* at $p < 0.05$. **c** NSS scaled between single observers and fixation maps' NSS per age bin. Black diamonds represent average deviations alongside 95% confidence intervals. $n = 1,600$.

**Model performance across individuals**. Models performed significantly better at predicting fixation locations of women ($n = 710$) than of men ($n = 890$; $BF_{10} = 244.904$, $t(20) = 4.779$, $p < 0.001$, Cohen's $d = 1.043$; see Fig. 2, Supplementary Figure 2 for spatial distribution maps of fixation locations for men and women, and Supplementary Table 3; for results on participants who reported other gender, see Supplementary Table 4 and Supplementary Figure 4). The average difference in predictions across gender for all models (NSS = 0.017) closely corresponded to the difference for the central bias (NSS = 0.018, Supplementary Table 3).

Further differences in model performance were found across age groups. Data were binned in nine groups, each spanning 6 years of age from 6-59, which ensured at least 34 participants per bin (see Table 1 for demographics per bin). Model performance was then averaged across age bin averages to account for class imbalances in the grand average, and this was used as baseline for across-age comparisons. Therefore this average, unlike the values reported in Table 2, is not biased towards the groups with most participants. Age biases across models were fairly consistent (Fig. 2; Table 3, see Table 4 for $t$-tests and respective effect sizes between bins). One age group revealed the clearest positive

deviation from average model performance: Those of the arguably most oversampled population, corresponding to most college students and young academics, respectively (18-29). This was accompanied by generally less consistent or worse predictions for other age bins. Note that, due to smaller group sizes, the age bins 6–11 and 54–59 warrant more caution before interpretation than

**Table 1 Demographics per age bin**

| Bin | Sample size | Age Mean | Gender Men (%) | Women (%) |
|-----|-------------|----------|----------------|-----------|
| 6–11 | 58 | 9.97 | 48.3 | 51.7 |
| 12–17 | 149 | 14.28 | 64.4 | 35.6 |
| 18–23 | 249 | 20.44 | 55.4 | 44.6 |
| 24–29 | 431 | 26.10 | 54.5 | 45.5 |
| 30–35 | 242 | 32.26 | 50 | 50 |
| 36–41 | 164 | 38.60 | 51.2 | 48.8 |
| 42–47 | 175 | 44.36 | 60.6 | 39.4 |
| 48–53 | 97 | 50.08 | 55.7 | 44.3 |
| 54–59 | 34 | 55.94 | 82.4 | 17.6 |

**Table 2 Performance of visual saliency models and baselines.**

| Model | NSS | Authors | Improvement (%) |
|---|---|---|---|
| Baselines | | | |
| Fixation map | 0.709 | | 100.00 |
| Central bias | 0.014 | Tatler[14] | 0.00 |
| Single observer | 0.176 | | 23.32 |
| Meaning map | 0.382 | Henderson & Hayes[15] | 52.97 |
| Models | | | |
| SALICON | 0.462 | Jiang et al.[21] | 64.45 |
| SalGAN | 0.434 | Pan et al.[22] | 60.42 |
| DeepGazeIIE | 0.411 | Linardos et al.[23] | 57.15 |
| DeepGazeII | 0.408 | Kümmerer et al.[35] | 56.78 |
| QSS | 0.350 | Schauerte & Stiefelhagen[45] | 48.37 |
| IMSIG | 0.342 | Hou et al.[46] | 47.20 |
| DeepGazeI | 0.339 | Kümmerer et al.[47] | 46.83 |
| DVA | 0.307 | Hou & Zhang[48] | 42.13 |
| SSR | 0.280 | Seo & Milanfar[49] | 38.35 |
| SAM | 0.279 | Cornia et al.[20] | 38.10 |
| ICF | 0.269 | Kümmerer et al.[50] | 36.70 |
| AIM | 0.255 | Bruce & Tsotsos[51] | 34.75 |
| IKN | 0.206 | Itti et al.[1] | 27.66 |
| RARE2012 | 0.200 | Riche et al.[52] | 26.79 |
| BMS | 0.194 | Zhang & Sclaroff[53] | 25.88 |
| CAS | 0.172 | Goferman et al.[54] | 22.71 |
| GBVS | 0.171 | Harel et al.[55] | 22.65 |
| SUN | 0.166 | Zhang et al.[56] | 21.85 |
| FES | 0.060 | Rezazadegan Tavakoli et al.[57] | 6.72 |
| LDS | 0.043 | Fang et al.[58] | 4.29 |
| CVS | −0.076 | Erdem & Erdem[59] | −12.86 |

Model performance (NSS, Normalized Scanpath Saliency score) for baselines and models are given as rows. Negative numbers denote worse than chance performance. The Improvement column denotes relative performance between central bias (0%) and fixation map (100%) in % NSS.

the other age bins. Notably, variation in model performance was relatively large for under-aged (<18 year-old) participants compared to other participants. Again, the central bias showed a largely similar tendency to the overall model biases, in line with findings reported earlier on a smaller selection of models and age groups[18,19]. Spatial distribution maps of fixation locations are given per age bin in Supplementary Figure 2; absolute NSS per model and baselines are given in Supplementary Table 1. Individual predicted maps per model (Supplementary Fig. 6), and deviations between models' predictions and actual fixation locations (Supplementary Fig. 7) are given in the supplementary information. The degree of biases across age became even more apparent with NSS scaled between single observers and fixation maps' NSS per age bin (Fig. 2c). Differences were striking here, again with best performances for participants in young adulthood (Fig. 2). Differences as a function of age were substantial ($F(1.699, 33.971) = 38.269$, $p < 0.001$, $\eta^2 = 0.657$) and were highly significant between most age bins. For instance, performance for children aged 6–11 differed from performance of all adults (all $p < 0.001$) except those in the oldest bin (see Table 4 for full post-hoc $t$-tests).

**Model performance across fixations**. Model performance was characterized by substantial variation in NSS, but revealed that fixations are predicted differently well over time. Here, the fixation map, reflecting an upper bound, showed much higher NSS scores early than later on, likely because fixations were much more focal for early viewing (see Fig. 3d). Models predicted the subset of early fixations generally better than when consecutive fixations were added, but only in absolute NSS scores (Fig. 3a; e.g., first vs. ninth fixation $t(20) = 5.881$, $p < 0.001$, 95% CI = [0.692, 1.857]). When scaled to the maximum achievable NSS, many models improved relative to this maximum as consecutive fixations were added, given that the NSS scores of the fixation map and central bias also decreased (Fig. 3b, first vs. ninth fixation $t(20) = -6.634$, $p < 0.001$, 95% CI = [−2.057, −0.821]). Relative performances across models differed as a function of the cumulative fixations made - e.g., SAM[20] performed very well on the first two fixations in comparison to the overall benchmark, which was best predicted by current, leading deep learning models such as SALICON, SalGAN, and DeepGazeIIE[21–23]. This variability showed in different model rank orders between the first few fixations, relatively later fixations, or all 18 fixations, respectively (corresponding to the leftmost and the rightmost rank order in Fig. 3b). Of course, a map of cumulative fixations contains more information whereas a map for only the first fixation is arguably more sparse - despite already 2607 fixations contributing to it. Are differences in performance across fixations therefore driven by data sparsity? Fixation maps for only the first and the ninth fixation, respectively, (Fig. 3d) compared with the overall map of all fixations (Fig. 1), showed markedly different patterns as function of early- versus late viewing behavior: Fixation maps were highly focal for the first and much more spread out for later fixations, data sparsity cannot have driven these effects. The benchmark of models per fixation (Fig. 3c) further showed relatively worse performance for early fixations (first vs. ninth fixation $t(20) = -4.679$, $p < 0.001$, 95% CI = [−1.544, −0.482]). Around six to ten fixations marked the point of best performance, which roughly matches the overall number of fixation clusters observed. NSS scaled between central bias and fixation map then dropped again across models. This challenges the current approach of optimizing saliency models on just one fixation map per image - in fact, what attracts gaze early on might be substantially different from what attracts gaze after a few fixations or extended viewing.

## Discussion

Here we set out to investigate how well saliency maps, models that have been proposed to predict the deployment of visual attention, and by extension fixation locations, generalize across individuals and the number of fixations. Using a sample of 2607 participants and 21 highly influential saliency models, gender and age biases in model performance were found for the subset of 1600 participants with credible demographics. Specifically, predictions were better for women and adults aged 18–29. These demographics (women, 18-29), perhaps incidentally, best represent those of the majority of participants in psychological research in general (predominantly younger adult women), as well as in the vast majority of benchmarking data[4,10]. Overall, a large portion of the biases in predictions across demographic groups followed relative differences in prediction of the central bias baseline, in line with previous work with smaller samples and more distinct age groups[18,19,24,25].

Besides demographic-based differences, model performance differed as a function of which fixation was to be predicted. Here, models only performed well for early fixations in absolute NSS, but actually worse when scaled with the maximum achievable NSS. Our sample allowed us to study how model performance evolves as a function of the number of fixations, as maps are not sparse; even if only the first or ninth fixation is used as a basis, it contains information from 2607 fixations. Worse predictions for early and later fixations compared to intermediate (i.e., sixth to

**Table 3 Differences in model performance split by age groups.**

| | Deviation in model performance relative to average model performance across age bins | | | | | | | | |
|---|---|---|---|---|---|---|---|---|---|
| **Model** | **6–11** | **12–17** | **18–23** | **24–29** | **30–35** | **36–41** | **42–47** | **48–53** | **54–59** |
| Baselines | | | | | | | | | |
| Fixation map | 0.186 | 0.028 | 0.003 | −0.103 | −0.105 | −0.105 | −0.065 | −0.018 | 0.179 |
| Central bias | −0.031 | 0.007 | −0.02 | 0.033 | 0.014 | −0.006 | −0.029 | 0.039 | −0.008 |
| Single observer | 0.071 | 0.026 | 0.034 | −0.009 | −0.021 | −0.037 | −0.016 | −0.021 | −0.029 |
| Meaning map | −0.029 | −0.024 | 0.042 | 0.047 | 0.015 | −0.002 | −0.006 | −0.006 | −0.037 |
| Models | | | | | | | | | |
| RARE2012 | −0.019 | −0.035 | 0.003 | 0.044 | 0.015 | 0.006 | 0.003 | 0.034 | −0.053 |
| SalGAN | −0.007 | −0.068 | 0.069 | 0.057 | 0.004 | 0.006 | 0.024 | 0.006 | −0.087 |
| DeepGazeIIE | 0.004 | 0 | 0.048 | 0.043 | 0.009 | −0.031 | −0.014 | 0.015 | −0.072 |
| SALICON | −0.006 | −0.012 | 0.071 | 0.039 | 0.003 | −0.011 | 0.008 | 0.024 | −0.115 |
| −DVA | −0.006 | −0.061 | 0.028 | 0.054 | 0.019 | −0.008 | 0.02 | 0.015 | −0.06 |
| FES | 0.008 | −0.029 | −0.004 | 0.04 | 0.021 | −0.001 | −0.014 | 0.016 | −0.036 |
| QSS | 0.037 | −0.034 | 0.035 | 0.036 | 0.01 | −0.024 | 0.004 | 0.009 | −0.069 |
| SSR | 0.005 | −0.046 | 0.045 | 0.026 | 0.031 | 0.005 | −0.006 | 0.028 | −0.089 |
| CVS | 0.02 | 0.011 | −0.015 | 0.01 | 0.013 | −0.004 | −0.023 | 0.008 | −0.024 |
| IMSIG | 0.013 | −0.035 | 0.036 | 0.044 | 0.03 | −0.011 | 0.006 | 0.017 | −0.096 |
| LDS | 0.018 | 0.021 | 0 | 0.021 | 0.001 | −0.013 | −0.03 | 0.01 | −0.031 |
| ICF | 0.001 | 0.024 | 0.017 | 0.037 | 0.007 | −0.041 | −0.018 | 0.031 | −0.059 |
| GBVS | −0.022 | 0.015 | −0.005 | 0.051 | 0.001 | −0.014 | −0.026 | 0.044 | −0.043 |
| CAS | 0.023 | 0.017 | 0.004 | 0.026 | 0.007 | −0.019 | −0.02 | 0.033 | −0.068 |
| SUN | 0.018 | −0.018 | −0.004 | 0.018 | 0.011 | −0.024 | −0.018 | 0.041 | −0.021 |
| DeepGazeI | 0.003 | −0.008 | 0.044 | 0.046 | 0.009 | −0.025 | −0.008 | 0.027 | −0.087 |
| AIM | 0.024 | 0.005 | 0.01 | 0.017 | 0 | −0.049 | −0.022 | 0.038 | −0.025 |
| SAM | −0.001 | 0.082 | 0.033 | 0.015 | −0.04 | −0.03 | −0.019 | 0.033 | −0.073 |
| DeepGazeII | −0.009 | −0.041 | 0.038 | 0.064 | 0.014 | −0.012 | 0.001 | 0.011 | −0.065 |
| IKN | −0.014 | −0.024 | 0.006 | 0.059 | 0.014 | −0.008 | −0.017 | 0.035 | −0.053 |
| BMS | 0.041 | 0.007 | 0.027 | 0.026 | 0.017 | −0.016 | −0.02 | 0.019 | −0.101 |
| Summary statistics (models only) | | | | | | | | | |
| Mean deviation | −0.006 | −0.011 | **0.023** | **0.037** | **0.009** | **−0.015** | **−0.009** | **0.024** | **−0.063** |
| Bayes Factor$_{10}$ | 0.739 | 0.567 | **92.727** | **>5000** | **6.982** | **258.279** | **4.050** | **>5,000** | **>5,000** |

Deviations (NSS) per age bin from the average across age bins. Negative numbers indicate worse model performance compared to other age bins, whereas positive numbers indicate better performance compared to other age bins. Summary statistics (excluding baselines) are given in the bottom two rows (bold: $p < 0.05$; inferential $t$-tests in Supplementary Table 2, tests for contrasts between age bins in Table 4).

tenth) fixations highlight the importance of more closely defining whether earlier or later viewing behavior is modeled. One possibility to account for the variability in prediction quality across fixations lies in adjusting how saliency models apply thresholds: For first fixations, only the few most focal locations could be emphasized by greedier thresholds to represent the focal distribution of fixation locations observed early on. For subsequent fixations, a more liberal approach might be employed, allowing more spread-out predictions - as observed for intermediate or later fixations. The here reported data further suggest that different models might capture different visuo-attentional processes - with some models being more predictive of early fixations [e.g.,[20]] and others being better at predicting later fixations [e.g.,[21,23]]. If the goal is to define which part of a scene is fixated first, particular models may perform best and thus be the method of choice. If the goal is to predict which parts of a scene will be fixated eventually, however, different models that weigh objects and semantic information more strongly are to be recommended. This view could resolve a number of outstanding debates on whether low-level features or semantics drive gaze behavior most strongly[5,15,26–29]. We speculate that both accounts have their merit: The answer could depend on the viewing duration and the number of objects in a scene. Remarkably, fixation maps were very focal for the first few fixations. As soon as the number of fixations approached the number of bigger objects in our scene, model performance did not notably change further when using a cumulative fixation map, indicating that participants eventually fixated most objects, but in differing sequences. Very late

fixations, in turn, could disperse even further and be captured worse as a consequence. Saliency models are optimized using cumulative fixation maps obtained from several seconds of free viewing. This practice might have introduced bias, as later fixations are disproportionately weighed in these maps relative to the initial two or three fixations. The findings and account put forward here could (partially) explain differences across benchmarking datasets and results[4]. A straightforward prediction would be that benchmarks that use gaze data on images with a short viewing duration favor models that prominently weigh relevant low-level features [possibly such that are common in faces[5,27]], whilst benchmarks with gaze data obtained from viewing behavior over a longer time favor models that weigh semantic content more strongly.

Taken together, the reported findings put the current approach of evaluating and improving models into question, which is predominantly to design and benchmark models around fixation maps constructed from several seconds of free viewing by college student participants. More generally, the present findings reveal that models of psychological processes - even as fundamental as low-level visual behavior - can be affected by systematic and substantial biases introduced via training and benchmarking datasets. Proper modeling and understanding of human spatial gaze behavior will require an approach that incorporates diverse samples, if the aim of said models is to predict behavior of more than just college students. Furthermore, saliency models could improve further by addressing effects of early versus later fixations, or this issue should be explicitly addressed in limitations of

**Table 4 Contrasts for percentage explained NSS of the explainable NSS between all age bins.**

| Comparison | | Mean difference | t | p (Bonferroni) | 95% CI 95% CI (Lower) | 95% CI (Upper) |
|---|---|---|---|---|---|---|
| 6–11 | 12–17 | −3.889 | −2.548 | 0.424 | −8.854 | 1.077 |
| | 18–23 | −8.191 | −5.367 | <0.001 | −13.157 | −3.225 |
| | 24–29 | −19.075 | −12.499 | <0.001 | −24.041 | −14.11 |
| | 30–35 | −16.039 | −10.509 | <0.001 | −21.004 | −11.073 |
| | 36–41 | −14.151 | −9.272 | <0.001 | −19.117 | −9.185 |
| | 42–47 | −11.345 | −7.434 | <0.001 | −16.311 | −6.379 |
| | 48–53 | −16.375 | −10.73 | <0.001 | −21.341 | −11.409 |
| | 54–59 | −3.811 | −2.497 | 0.487 | −8.777 | 1.154 |
| 12–17 | 18–23 | −4.302 | −2.819 | 0.195 | −9.268 | 0.663 |
| | 24–29 | −15.187 | −9.951 | <0.001 | −20.153 | −10.221 |
| | 30–35 | −12.15 | −7.961 | <0.001 | −17.116 | −7.184 |
| | 36–41 | −10.262 | −6.724 | <0.001 | −15.228 | −5.296 |
| | 42–47 | −7.457 | −4.886 | <0.001 | −12.423 | −2.491 |
| | 48–53 | −12.487 | −8.182 | <0.001 | −17.452 | −7.521 |
| | 54–59 | 0.077 | 0.051 | 1 | −4.889 | 5.043 |
| 18–23 | 24–29 | −10.885 | −7.132 | <0.001 | −15.85 | −5.919 |
| | 30–35 | −7.848 | −5.142 | < 0.001 | −12.813 | −2.882 |
| | 36–41 | −5.96 | −3.905 | 0.005 | −10.926 | −0.994 |
| | 42–47 | −3.154 | −2.067 | 1 | −8.12 | 1.811 |
| | 48–53 | −8.184 | −5.363 | <0.001 | −13.15 | −3.218 |
| | 54–59 | 4.379 | 2.87 | 0.168 | −0.586 | 9.345 |
| 24–29 | 30–35 | 3.037 | 1.99 | 1 | −1.929 | 8.003 |
| | 36–41 | 4.925 | 3.227 | 0.055 | −0.041 | 9.891 |
| | 42–47 | 7.73 | 5.065 | <0.001 | 2.764 | 12.696 |
| | 48–53 | 2.7 | 1.769 | 1 | −2.265 | 7.666 |
| | 54–59 | 15.264 | 10.002 | <0.001 | 10.298 | 20.23 |
| 30–35 | 36–41 | 1.888 | 1.237 | 1 | −3.078 | 6.854 |
| | 42–47 | 4.693 | 3.075 | 0.089 | −0.273 | 9.659 |
| | 48–53 | −0.337 | −0.221 | 1 | −5.302 | 4.629 |
| | 54–59 | 12.227 | 8.012 | <0.001 | 7.261 | 17.193 |
| 36–41 | 42–47 | 2.805 | 1.838 | 1 | −2.16 | 7.771 |
| | 48–53 | −2.224 | −1.458 | 1 | −7.19 | 2.741 |
| | 54–59 | 10.339 | 6.775 | <0.001 | 5.373 | 15.305 |
| 42–47 | 48–53 | −5.03 | −3.296 | 0.044 | −9.996 | −0.064 |
| | 54–59 | 7.534 | 4.937 | <0.001 | 2.568 | 12.5 |
| 48–53 | 54–59 | 12.564 | 8.233 | <0.001 | 7.598 | 17.529 |

Contrasts are for data as given in Fig. 2c and Table 3 across age bins. Data distribution was assumed to be normal but this was not formally tested for these contrasts.

models. Different models might be used to account for these differences or models could incorporate adjustable options: For whom and for when should fixations be predicted? Knowledge about developmental differences in sampling behavior and saliency computations[18,25,30–32] could be incorporated here, as well as findings into differences in fixations across viewing duration [e.g., refs. [33,34]].

Most models outperformed the central bias, in line with several benchmarking results[4]. However, many models already incorporate a central bias and the image at hand features many spread-out objects. Generally, more recent models performed best at predicting the overall fixation map (i.e., including data of all participants and fixations), some of which are currently also leading in benchmarks across stimuli [e.g., ref. [23]]. Indeed, deep learning-based models[21–23,35] generally outperformed more traditional and interpretable models which are foremost centered around low-level image feature computations. However, these deep-learning models still suffered from qualitatively similar limitations regarding generalizability across individuals and fixations. Furthermore, the meaning map was outperformed by several models, in contrast to initial findings[15], reminiscent of recent criticism[29], but see Henderson et al.[26]. Meaning maps could possibly be advanced further by ensuring high

correspondence between the demographics of raters and participants whose fixations are to be predicted.

**Limitations**. Naturally, the sample put forward here, while vast, is in turn affected by sampling biases. For example, children need to be willing to wait and focus, older adults need to have sufficient vision to participate. Which further differences between people, beyond gender and age, are relevant when it comes to saliency maps remains to be determined. With this first step, we hope to stimulate research into this, including into non-western populations in the field of saliency mapping, as suggested in many areas of psychology[11,13,36]. As a way forward, to overcome the biases uncovered here, we provide the present database to help improve across-participant and across-fixation generalizations and will supply it with additional data as long as the exhibition remains active. Replications of presently reported findings across multiple images would be desirable, for instance using other large scale data collections such as described here. Other limitations to be kept in mind lie in the less controlled setup than used in most common benchmarking data sets, or limited possible inferences on gaze behavior of participants with non-binary gender. The practical implications here are simple: no gender should be made

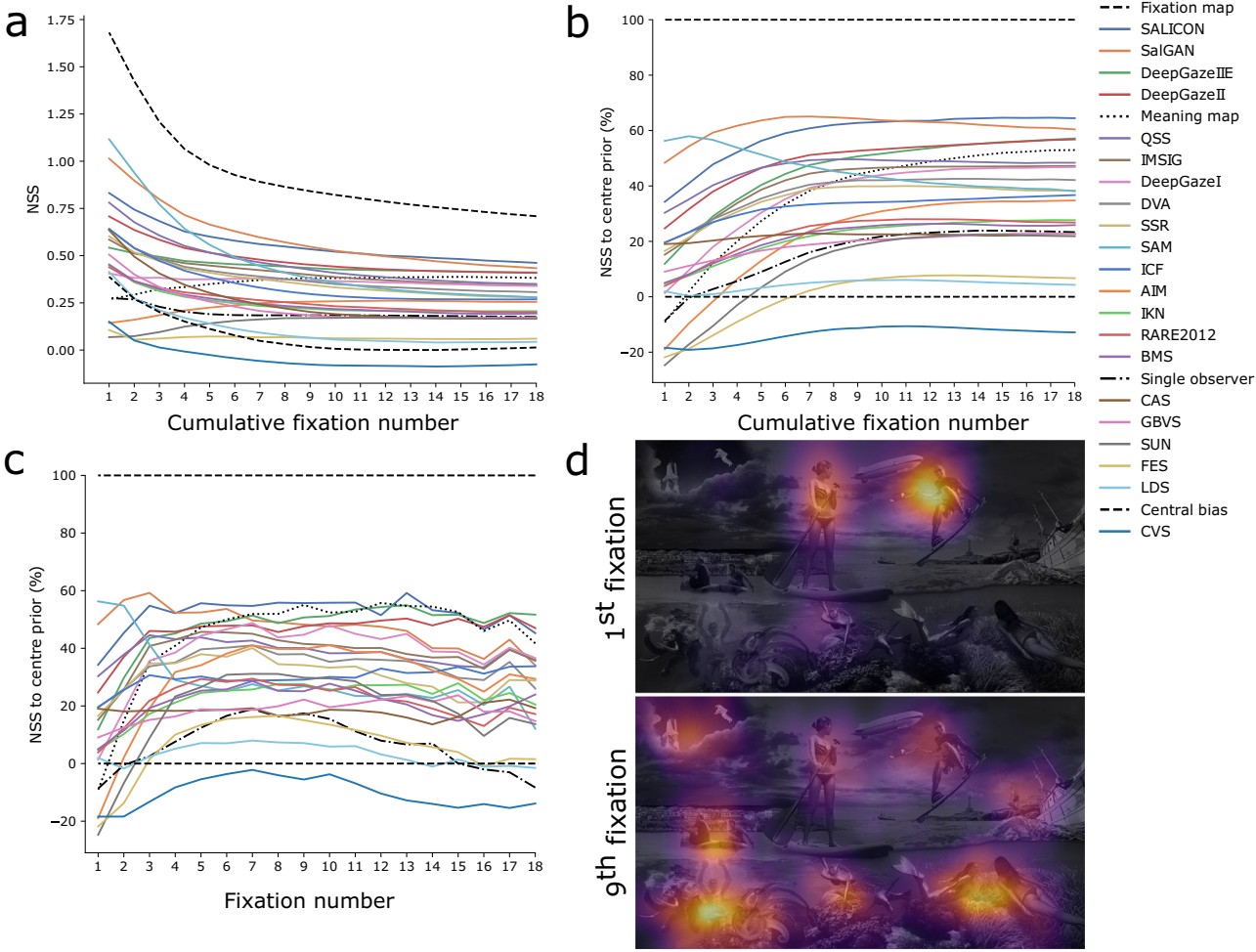

**Fig. 3 Model performances across fixations. a** Model (colors) and baseline (dashed/hatched/dotted) benchmarking across fixations with unscaled NSS. **b** Model benchmarking across fixations with NSS scaled to maximally achievable NSS per fixation. Fixation maps are cumulative, i.e., righter points on the x-axis indicate model performance including all previous fixation data. The rightmost data corresponds to the benchmarking reported in Table 2. **c** Same visualization as in b, but per fixation instead of cumulative fixations. **d** Fixation map for only the 1st fixation shows a much more focal distribution of fixations than for the 9th fixation, which much more closely resembles the map after all 18 fixations (Fig. 1 right). $n = 2{,}607$.

the default option and 'prefer not to say' options should be given to not obscure findings. Whilst we did not find qualitative differences in findings for two different smoothing kernel sizes to create the fixation maps, more extreme choices for kernels or flexible central biases might affect the here reported bounds and thus findings.

**Conclusions**. Past findings [e.g., on the central bias outperforming many models[14]] could have been taken to cast doubt on the general usefulness of saliency modeling. Such criticisms, however, have led to even increased efforts to develop more powerful models, e.g., by incorporating the central bias. The here identified systematic challenges to saliency modeling - generalizability across individuals and which fixation is to be predicted - might require new approaches in turn, for instance by incorporating information besides low-level image features [see also ref. [37] for related calls] in order to make the next leap forward happen. If models can only generalize to overall fixation maps across images[4], but fail at generalizing across earliest fixations or work best only for certain demographic samples, different models might be needed to predict different (groups of) individuals and visuo-attentional processes. Or, even better, models might incorporate which fixations are to be predicted - distinctions for the demographics and the number of the fixation could hereby set

the path for much more powerful models. We argue that visual saliency should be considered a dynamic, interactive, and integrative result of low-level image features [e.g.,[1]], as well as semantic information [e.g., refs. [5,37]] or meaning maps[15] under the consideration of the individual differences that have been associated with gaze behavior [e.g., refs. [30,31]].

## Methods
The study was approved by the Ethics Review Board of the Faculty of Social Sciences at Utrecht University.

**Participants and data exclusion**. Overall, $n = 2{,}607$ valid free viewing gaze data sets were obtained, using an installation at the NEMO Science Museum, Amsterdam, which featured a metal box with a screen and an eye tracker inside which participants looked into. All analyses not related to demographics were performed on all 2607 data sets ($M_{Age} = 28.79$, men = 50.13%, women = 42.9%; non-binary = 6.97%). For analyses relating to demographics, data sets were only considered if no periods of more than 5 s of lost gaze position were recorded over the duration of the whole procedure (including entering demographics). For data sets adhering to this requirement, it is highly unlikely that the participants left the recording between free viewing and entering their demographics. Data sets were further excluded from any

demographic-linked analyses if the default options (non-binary gender, year 2000 as year of birth) were not changed by the participant, resulting in $n = 1600$ participants with demographics of high credibility ($M_{Age} = 29.82$, men = 55.6%, women = 44.4%; see Table 1 for detailed demographic information). $N = 91$ participants indicated non-binary gender across the 6-59 age range, but given that there was no option for 'prefer not to say', these data are only given in the Supplement and have to be interpreted with caution.

**Statistics and reproducibility**. All statistical tests reported were two-sided. Tests are detailed alongside results. Bayesian tests use default JASP priors. The study was not preregistered.

**Apparatus, stimuli, and procedure**. Gaze was logged (asynchronously) at 60 Hz using a Tobii Eye Tracker 4C. This eye tracker is suited for this research question and setup. In general however, as the Tobii 4C filters the data for its intended use case (gaze interaction), Tobii advises against using it for research. A 27", 1920 × 1080 px monitor with a maximum luminance of 300 cd/m² was used for stimulus presentation, located at 80 cm distance from the eyes to the screen (50 × 24 degrees visual angle). A metal box around screen and tracker shielded the field of view from other visual stimulation. Participants could either stand, sit on a chair, or stand on a chair to be able to see the monitor and participate. Other than that, the setup was not height adjustable. Auditory information was given exclusively after free viewing via two loudspeakers positioned close to the participants' ears. See Supplementary Figure 1 for pictures of the setup.

Participants were required to look at a central circle that gradually filled to start the experiment and perform a five-point calibration of the eye tracker. Participants were presented with a full-screen collage image (Fig. 1) for 10 s of free viewing without instruction. The image was constructed so that it would include a wide variety of objects, both inanimate and animate, both facing or not facing the beholder, as well as free spaces with low information (i.e., empty sea or sky). Participants could decide on whether or not to donate their data by fixating a laterally positioned 'yes' or 'no' button respectively.

Upon giving consent, participants were prompted to indicate their gender by gazing at a central (non-binary), left (man), or right (woman) circle. Subsequently, year of birth could be entered, with 2000 as default option. This year could be iteratively decreased or increased by gazing at a circle on the left or on the right, respectively.

**Data quality and preprocessing**. Eye tracking data quality can be assessed by precision, accuracy, and data loss[38,39]. While accuracy cannot be assessed with the current setup, precision, calculated as in Hooge et al.[40], was $Mdn = 0.68°$ (SD = 0.28°), loss was $M = 0.8\%$ (SD = 2.4%). These are reasonable values given the special nature of the setup (see SI Data quality for more information and Supplementary Figure 5). Neither precision nor loss have visibly driven results across demographic groups. Fixation candidates were detected from raw gaze data with an algorithm specifically built for noisy data[41]. Fixation candidates were discarded if shorter than 60 ms, or merged if intermittent saccade candidates were smaller than 1 degree of visual angle in amplitude. This procedure has been demonstrated to prevent event-detection related biases[42]. Given that participants needed to look at the center of the screen to start free-viewing, all fixations with onsets before the start of free-viewing were removed from the dataset. The following eighteen fixations were considered for analyses to account for differential fixation counts of participants (this equated to $M = 6.906$ s of free viewing). Participants with

fewer than eighteen fixations ($n = 117$), thus deviating more than 1.5 median absolute deviation from the median, were excluded resulting in the total of 2607 participants (Number of fixations per participant: $Mdn = 25.0$ $MAD = 4.45$). Fixations that were located outside of the bounds of the screen of the experimental setup were excluded.

**Baseline spatial distribution maps**. To evaluate the performance of the predictions obtained from 21 saliency models tested here, four different baselines were constructed. (1) A map of all actual fixation locations served as the upper bound for the performance of any model (comparison between the binary array of fixation locations and its smoothed counterpart). This *fixation map* was constructed from fixation locations by applying a Gaussian filter (SD = 1 degree of visual angle) to the fixation map, effectively making it continuous[17,43] - in other words, discrete fixation locations were blurred over with this kernel. This approach allows to construct regions rather than pixels for fixation determination and acts as regularization for potential small scale measurement error[17]. (2) A *meaning-map*, a model created from successive ratings of small patches of the image by $n = 59$ participants served as gold standard model[15], possibly best incorporating information about objects and semantics, as previously proposed for computational models[5]. To create the meaning map, the image was split into overlapping patches with diameters of 1.5, 3 and 7 degrees of visual angle. These patches were then rated for meaningfulness by $n = 59$ participants ($Mdn_{Age} = 25$ years, SD = 7.399 years; men: 39, women: 19, non-binary: (2), recruited via Prolific without restriction regarding demographics, using Gorilla in an online experiment. This experiment took about 15 minutes to complete during which participants had to rate 200 patches each. Participants were each rewarded with 9 euros. (3) A *Gaussian central bias*[14], skewed to the aspect ratio of the screen (SD = screen half dimensions), served as the baseline performance that should be achieved by any model. Effectively, the central bias lets saliency be maximal at the center with declining saliency towards the edges of the screen. Central biases can outperform saliency maps[14] and are therefore incorporated in many of the more recent, here evaluated, models [e.g.,[23,35]]. (4) Lastly, a *single observer model* expressed how well one participant's gaze behavior matched gaze behavior of all other participants. This procedure was repeated over all participants (similar to leave-one-out cross-validation), and scores from all iterations were then averaged.

**Evaluation metric**. A multitude of evaluation metrics for saliency maps have been put forward [see refs.[17,43], for reviews] of which Normalized Scanpath Saliency (NSS) was used here. NSS correlates highly to other metrics and has generally favorable properties as it requires minimal prior assumptions[16,17]. NSS was extracted per model by first z-standardizing the respective saliency map, and overlaying it with the binary map of discrete fixation locations. For each fixated pixel, the z-score of the corresponding pixel was taken from the saliency map and a grand mean was calculated over those values. All maps (baselines or model predictions) were evaluated against the discrete fixation locations. For the single observer model, discrete fixation locations of one participant were evaluated against the blurred fixation map of all other participants. As such, NSS accounts for the relative saliency of regions as predicted by a given saliency map, not absolute saliencies that differ between models[17,44]. False positives and false negatives are equally weighed, and (nonbound) positive NSS scores indicate above chance-level performance, whereas negative NSS scores indicate worse than chance performance. As NSS is reduced to one score, it does not indicate which

regions drive better or worse than chance performance. For this reason, graphical representations of the delta between predicted models and the spatial distribution map of fixation locations are given in Supplementary Figs. 6 and 7.

**Reporting summary.** Further information on research design is available in the Nature Portfolio Reporting Summary linked to this article.

## Data availability

All code and data are available via the Open Science Framework https://osf.io/sk4fr/, https://doi.org/10.17605/OSF.IO/SK4FR.

## Code availability

All code and data are available via the Open Science Framework https://osf.io/sk4fr/, https://doi.org/10.17605/OSF.IO/SK4FR.

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

## Acknowledgements

We thank the NEMO Museum Amsterdam for their help with data collection and all participants who donated their data. We thank Tobii for consulting over data quality. This project has received funding from the European Research Council (ERC) under the European Union's Horizon 2020 research and innovation programme (grant agreement n° 863732). This research was funded by a VICI Grant 211-011 from the Netherlands Organization for Scientific Research to Stefan Van der Stigchel. The funders had no role in study design, data collection, and analysis, decision to publish or preparation of the manuscript.

## Author contributions

C.S. and S. v.d.S. conceptualized the studies. A.J.H., G.B., and C.S. conducted the analyses with help by I.T.C.H. and S.M.S., C.S. and A.J.H. wrote the draft of the manuscript. A.J.H. and C.S. performed the visualization of results. All authors (including T.C.W.N.) provided critical review & editing.

## Competing interests

The authors declare no competing interests.
