## [Peer Review File · Communications Psychology]

17th Jan 23

Dear Dr Strauch,

Thank you for your patience during the peer-review process. Your manuscript titled "None to rule them all? Systematic shortcomings in modeling saliency across individuals" has now been evaluated by 3 reviewers, and I include their very constructive comments at the end of this letter.

You will see that they find your work of some potential interest. However, they have raised quite substantial concerns that must be addressed. In light of these comments, we cannot accept the manuscript for publication, but would be interested in considering a revised version that fully addresses these serious concerns.

We hope you will find the Reviewers' comments useful as you decide how to proceed. Should additional work allow you to address these criticisms, we would be happy to look at a substantially revised manuscript. If you choose to take up this option, please highlight all changes in the manuscript text file, and provide a detailed point-by-point reply to the reviewers.

Editorially, we consider it necessary that all the concerns raised by the reviewers pertaining to data quality and to the effects of potential confounds on the strength of your interpretation must be addressed. Two issues to pay particular attention to are: (1) the potential difference in the quality of the data between the demographic variable of interest, in particular between age bins and (2) the differences in the data variance linked to variations in the sample size between each bins. Reviewers described in detailed a series of analyses that could mitigate the effects of these potentially confounding variable on the demographic effects. We believe that addressing these two points is necessary to ensure the interpretability of the findings reported in the manuscript.

Please use the following link to submit your revised manuscript, point-by-point response to the referees' comments (which should be in a separate document to any cover letter) and the completed checklist:

[link redacted]

If the revision process takes significantly longer than five months, we will be happy to reconsider your paper at a later date, provided it still presents a significant contribution to the literature at that stage.

Please use the following link to submit your revised manuscript, point-by-point response to the Reviewers' comments with a list of your changes to the manuscript text (which should be in a

separate document to any cover letter) and any completed checklist:

Please do not hesitate to contact me if you have any questions or would like to discuss the required revisions further. Thank you for the opportunity to review your work.

Best regards,

Eva R. Pool

Eva R. Pool, PhD
Editorial Board Member
Communications Psychology
orcid.org/0000-0001-5929-1007

EDITORIAL POLICIES AND FORMATTING

Editorial Policy: [Policy requirements](https://www.nature.com/documents/nr-editorial-policy-checklist.pdf) (Download the link to your computer as a PDF.)

Furthermore, please align your manuscript with our format requirements, which are summarized on the following checklist:

[Communications Psychology formatting checklist](https://www.nature.com/documents/commsj-psychol-style-formatting-checklist-article.pdf)

and also in our style and formatting guide [Communications Psychology formatting guide](https://www.nature.com/documents/commsj-psychol-style-formatting-guide-accept.pdf) .

*** TRANSPARENT PEER REVIEW:** Communications Psychology uses a transparent peer review system. This means that we publish the editorial decision letters including Reviewers' comments to the authors and the author rebuttal letters online as a supplementary peer review file. However, on author request, confidential information and data can be removed from the published reviewer reports and rebuttal letters prior to publication. If your manuscript has been previously reviewed at another journal, those Reviewers' comments would not form part of the published peer review file.

*** CODE AVAILABILITY:** All Communications Psychology manuscripts must include a section titled "Code Availability" at the end of the methods section. In the event of publication, we require that the custom analysis code supporting your conclusions is made available in a publicly accessible repository; at publication, we ask you to choose a repository that provides a DOI for the code; the link to the repository and the DOI will need to be included in the Code Availability statement. Publication as Supplementary Information will not suffice. We ask you to prepare code at this stage, to avoid delays later on in the process.

*** DATA AVAILABILITY:**

All Communications Psychology manuscripts must include a section titled "Data Availability" at the end of the Methods section or main text (if no Methods). More information on this policy, is available at <http://www.nature.com/authors/policies/data/data-availability-statements-data-citations.pdf>.

At a minimum the Data availability statement must explain how the data can be obtained and whether there are any restrictions on data sharing. Communications Psychology strongly endorses open sharing of data. If you do make your data openly available, please include in the statement:

We recommend submitting the data to discipline-specific, community-recognized repositories, where possible and a list of recommended repositories is provided at <http://www.nature.com/sdata/policies/repositories>.

If a community resource is unavailable, data can be submitted to generalist repositories such as [figshare](https://figshare.com/) or [Dryad Digital Repository](http://datadryad.org/). Please provide a unique identifier for the data (for example a DOI or a permanent URL) in the data availability statement, if possible. If the repository does not provide identifiers, we encourage authors to supply the search terms that will return the data. For data that have been obtained from publicly available sources, please provide a URL and the specific data product name in the data availability statement. Data with a DOI should be further cited in the methods reference section.

REVIEWER EXPERTISE:

All reviewers have expertise in vision science (incl eyetracking and gaze analysis) and computational modelling,

REVIEWERS' COMMENTS:

Reviewer #1 (Remarks to the Author):

SUMMARY

The paper presents an exceptionally large eyetracking dataset, recording the fixations of over 8,000 people, from small children to older adults, as they freely look at a single image. The authors then evaluate the performance of a large range of older and newer saliency models at predicting these fixations. The dataset presents an unusual opportunity to assess saliency models in depth against a

much broader demographic sample than existing benchmarks do. As such, I think the work is promising, although I believe further analyses are needed before the data can be meaningfully interpreted, as explained below.

MAIN COMMENTS

Given the framing of the paper around "modeling saliency across individuals" I was surprised to see that all of the analyses are performed at a group level. If I have understood correctly, model performance is always evaluated in terms of the Normalised Scanpath Saliency score (NSS) between the model's predicted continuous saliency map and a human saliency map that is derived by (Step 1) aggregating many participants' measured fixations together in a single map, then (Step 2) applying a gaussian blur to make these discrete fixations into a continuous map. Three different analyses are reported, which differ in how participants are aggregated at Step 1: Table 1 reports model evaluations when all 8,235 participants are included; Table 2 reports model evaluations when participants are grouped into nine bins by age; and Supplementary Table 3 reports the same when participants are grouped into two bins by gender. The paper's main conclusions hinge on differences in model performances across the nine age bins. However, differences in the sample size and likely data quality between bins make it difficult to interpret model performance differences.

As a crucial first step, we need to see the baselines of Spatial Distribution Map (SDM) and Single Observer (SO) *within each age group* (i.e. what is the NSS of the bin-averaged spatial distribution map? And what is the average NSS when using each individual within that age group to predict the bin-averaged spatial distribution map?). These provide us with, respectively, the maximum possible performance of any model within that bin, and a proxy for the amount of inter-individual variability within that bin (which gives an estimate of how well a model would perform if it were as good as another representative human individual). The peak performance of the best model within any age group appears to be 0.081 (QSS, 24-29yo, Table 2), which is substantially less than the performance of the best model when assessed using all of the data (0.226, Table 1). This makes sense, because the age-binned spatial distribution maps are likely noisier since they are created using far smaller sample sizes (max bin N = 338) than the all-participant data (N = 8,235). Most importantly, the noisiness of the data almost certainly differs between age bins, both because sample sizes vary (from N = 36 to N = 338) and because participants of different ages likely give differently noisy data (e.g. young children may look for shorter durations, or have greater head motion). These are inevitable imperfections in any such large dataset gathered "in the wild" and are not necessarily a problem. *However* they mean that to meaningfully interpret the data, the baseline/ceiling values (for SDM and SO) should be shown separately for each age group, and ideally model performances should be expressed as a proportion of the explainable "signal" available in the data (e.g. model NSS divided by SO NSS, within each bin).

It would also be good to mention why a group analysis is necessary at all - if interested in predicting gaze in individuals, why not measure each model's NSS against each individual's blurred fixation map (and using the group-average SDM to estimate the ceiling of possible model performance)? Presumably because the sparseness of fixations and high variation across individuals renders such an analysis very noisy?

MINOR COMMENTS

- It would be good to provide a bit more detail in the Introduction on some of the most widely-used

saliency benchmark datasets: How many images and participants? Does each participant look at each image, or do the data for different images come from different participants? What is the age range of participants? Are models standardly evaluated against the participant-average fixation map for each image, or against each individual participant for each image? This information would help the reader appreciate the differences between common benchmarks and the current dataset.

- Briefly describe the data-collection context early in the paper (i.e. around Figure 1 in the Introduction). The fact that it was collected at a museum via an interactive exhibit helps the reader understand why just a single image was used, and how such a large sample size was obtained.

- Table 1: Provide more information in the table caption. e.g. briefly explain "NSS", what "% improvement" is relative to, and what "spatial distribution map" refers to.

- p11. It's stated that participants were excluded from demographic analyses if they didn't change the default settings (birth year = 2000, gender = nonbinary), but it's not clear whether participants were excluded if *either* of these was unchanged (which would systematically exclude all nonbinary people from analyses), or only if *both* of them were unchanged (which would exclude the much smaller subset of nonbinary people born in 2000).

- The proportion of participants who reported demographics is very low (1,412 out of 8,235 = 17%). This is understandable - participant commitment in an interactive museum exhibit is likely to be low, people may be reluctant to provide personal information, and the gaze-contingent method of reporting demographics (though potentially engaging) may be unweildy. However, it would be good to acknowledge this limitation in the discussion, and perhaps comment on whether certain age groups may have been more likely to provide demographic information than others, leading to systematic over-representation of some age groups in the no-information majority of the dataset.

- To the extent that findings are inconsistent with previous reports in the literature (e.g. overall poor performance of most models relative to the centre prior baseline), how much might this be due to the unusual nature of the single probe image? Most work on developing saliency models has considered "natural images" (i.e. human-taken photographs), which have a quite different spatial structure and distribution of "meaningful elements" than this digital collage image. It could even be argued that the use of such "unnatural" images is a strength, in that they can showcase aspects of gaze behaviour that are usually neglected in saliency research. However, it would be good to consider this point in the Discussion.

Reviewer #2 (Remarks to the Author):

This paper evaluates a range of salience models against the fixations of more than 8,000 observers, freely viewing a single image in a museum setting. Additionally, it tests whether model performance varies as a function of gender and age, using a subset of more than 1,400 observers who provided corresponding data.

In general, the performance of salience models was poor. Fewer than half of the tested models outperformed a simple centre prior. Notably, three models - including GBVS - outperformed a 'meaning map'. The best performing model (DeepGazellE) was far worse than the average performance for using a single observer's data to predict the fixations of all others.

There was a non-significant trend for better model performance for male vs female observers and significant variation across age bins. Model performance peaked for young adults and was particularly bad for children below the age of 12. The authors conclude that a narrow focus on college student populations runs the risk of myopic models and overfitting.

Strengths:

The presented dataset, to the best of my knowledge, is the largest of its kind. Processed fixation data is already available via OSF and the authors seem to plan to release the raw data as well, which would render this a highly valuable resource for the field.

Although previous studies compared free-viewing data of children and adults (including the fit of low-level salience; e.g. Acik et al., 2010 *Front Psychol*), the current sample allows to quantify the fit to salience models over an unprecedented range with reasonable granularity. It is particularly interesting that model performance peaks around the age range typically sampled in previous studies and drops off at either side of this. The authors' call for awareness of sampling biases and their drawbacks is well justified.

Given the large sample size and high power, the absence of significant gender differences is meaningful and interesting in its own right.

The observation that GBVS outperformed meaning maps is a relevant contradiction to previously published results.

Problems to be addressed:

Quality of fixation data

I'm unsure about the quality of fixation data. A median of 11 fixations appears extremely low for 10s viewing time – less than half of what would typically be expected. I suspect this is related to the fixation definition requiring a minimum gap of 60ms between successive fixations. Typical saccade durations are 20-40ms, so this seems a highly surprising parameter choice which may well lead to a loss of 50% of fixations. Without access to the raw data it is hard to judge what is going on, but a look at the processed fixation data reveals systematic gaps of hundreds of ms between successive fixations. Given all other analyses hinge on this, this point strikes me as crucial. Please double-check the procedure and provide example X,Y traces and their segmentation into events for a couple of observers and trials. If my suspicion is warranted, the raw data should be re-analysed using appropriate parameters, which may or may not change the pattern of results (e.g. if children make shorter saccades a higher fraction of fixations may have been dropped because of shorter inter-fixation gaps).

The surprisingly small number of fixations may also be down to excessive data loss. The authors claim that 'data loss cannot be assessed with the current setup' (p. 11), shortly after explaining that 'no periods of more than 5s of lost gaze position' (p.10) served as an exclusion criterion. This seems contradictory and is in need of explanation. The proportion of samples with lost gaze position should

be quantified and reported.

The authors further report precision estimates based on inter-sample distance. This rests on the assumption that successive samples are independent estimates of position. Having worked with the 4c, this seems unlikely. Tobii doesn't provide information about their proprietary filtering algorithms, but the position estimates appear visibly 'sticky', as in a moving average or Kalman filter. This of course would render inter-sample distance an inappropriate proxy to precision. As long as there's no evidence of successive samples being truly independent measures, this precision proxy should be avoided.

Finally, the authors used a single point calibration procedure and did not collect any validation data. At the same time the screen (and image? See below) covered a vast visual angle. I'm not overly worried regarding this point, because of personal experience with the 4c and the reasonable looks of the heatmaps. But it should be feasible to collect proper calibration and validation data for the full screen post-hoc. Data from just a few participants (ideally children and adults) would render the authors case for data quality much more compelling.

Single image

The use of a single image means we don't know how well the results generalize. This is compounded by the peculiar nature of the collage image, which renders it different from a single natural scene. The heatmaps appear clearly driven by the collage nature of the stimulus. This should be discussed.

***Unequal sampling ***

The number of data points per bin is extremely unequal between age groups and particularly low for children under 12. Could this be a confound? To be sure, the authors could randomly subsample an equal number of observers from the older bins and in each iteration calculate the NSS for the subsample. If the observed NSS for children is well outside the distribution of values observed for random adult subsamples of the same size, this confound can be excluded.

Unknown demographics for model comparisons

The comparison of model performances rests on data with unclear demographics. Therefore, it remains open which role sample specifics play for these general observations. For example, does GBVS outperform meaning maps in all age brackets? Supplementary analyses zooming in on the data with known demographics could shed light on this. A good way of showing this would be to provide a version of figure two colour coding all models (plus baselines) without obscuring them by the median line. Relatedly, for the youngest bin the median line seems to fall below the majority of models (bit hard to tell, but I count at least 12 lines ending above the median, which doesn't seem to add up?).

More details on methods, please

The methods section is very sparse and should convey more detail. For example: Did image size correspond to screen size? What kernel was used for smoothing fixation maps? Was a kernel used for NSS fixation sampling, or did a fixation correspond to a single pixel? What was the physical setup? Did observers stand or sit? How were size differences in children accounted for? What were the instructions given via the speakers? Also – are you sure about a sampling rate of 60 Hz – the default for the 4c is 90 Hz.

Previous developmental studies

Please relate your findings to those of previous developmental studies (you cite some) and demarcate the advance of the present study compared to them.

Methods contain a doubled sentence on highlighting of three most fixated sub-images.

Reviewer #3 (Remarks to the Author):

The manuscript by Strauch et al. investigates generalizations in saliency predictions across a very large number of individuals. Overall, model performance is well, however, breaks down for younger and older age groups than the most often studied group containing university students. These results are important and interesting for a wide audience.

The manuscript is clearly and succinctly written. The literature is adequately cited and the Methods seem sound. The team of author clearly represents high expertise. I have no major critique.

I wonder if the term "centre prior" could be misleading for some readers, since it might suggest Bayesian terminology, which is, however, not the case here. Therefore, I suggest to use "centre bias" instead.

The language should be checked with respect to use of English vs. American (e.g., no consistent use of "modeling" vs. "modelling").

EOF

Point by point response

We were happy to read the constructive points raised by the reviewers to improve our initial submission and the overall positive response to our initial work, especially on the usefulness of a vast and openly accessible eye-tracking data set. Reviewer 2, however, spotted an important issue in our data: the number of fixations was suspiciously low. Upon closer inspection, we realized that we had in fact been provided with a *filtered* gaze signal. Whilst informative about gaze position to a certain degree, we decided to not use the previous data, given that we could not get access to the filtering algorithm. Therefore, we collected new – raw and unfiltered – data using the same set-up of more than 2,500 participants – which caused the relatively long delay until we could revise the manuscript. We will continue supplying this data to the open science framework repository for as long as the installation remains active as a resource to the community. Importantly, however, the overall finding on biases across groups only changed partially and the most important conclusions still stand.

Furthermore, the higher resolution of the new data allowed to not only compare saliency models across age groups (which, due to a much lower data exclusion in the current data set can again be evaluated), but also across fixations – i.e., whether models perform better or worse for early or late fixations.

Unsurprisingly, these changes have resulted in a substantially revised manuscript (apart from the introduction). We apologize for the extra effort this causes in review and want to thank the reviewers for their helpful comments. We would like to explicitly thank Reviewer 2 for catching the major issue, this is peer-review as it should be and helped to substantially improve the manuscript.

To summarize: Because we now use the unfiltered data, the data is now much richer in detail on the number of fixations and provide a stronger test for the tested models. Models now clearly outperform the central bias and some models still outperform the meaning map. Regarding generalizability across participants, we now find a difference across gender (with better predictions for females) and age effects similar in quality to those reported initially, albeit less pronounced. Across age, gaze behavior of young adults (corresponding to most college students) was predicted best – with less consistent predictions for minors and worse predictions for older adults. Newly added analyses regarding generalizability across fixations further showed an intriguing finding: model benchmarks differ drastically, depending on whether early fixations (such as the 2nd or 3rd fixations) or later fixations (such as the 15th or 16th fixations) are to be predicted. These findings on generalizability highlight the importance of biases in training/testing data, but also the need to consider which gaze behavior and underlying visuo-attentional process is predicted and for whom. Together, these learnings have the potential to resolve a number of inconsistencies in saliency modeling literature (such as different benchmarking results on different data sets) and could pave the way to a new generation of much more powerful saliency models which are more sensitive to the properties of individuals and the underlying visuo-attentional process.

We addressed all raised comments by the reviewers below. We attached a version in which added/modified parts of the manuscript are highlighted in blue (except figure and references; removed parts are not indicated).

Green: Our answers

Blue: Edits in manuscript

Reviewer comments:

Reviewer #1 (Remarks to the Author):

SUMMARY

The paper presents an exceptionally large eyetracking dataset, recording the fixations of over 8,000 people, from small children to older adults, as they freely look at a single image. The authors then evaluate the performance of a large range of older and newer saliency models at predicting these fixations. The dataset presents an unusual opportunity to assess saliency models in depth against a much broader demographic sample than existing benchmarks do. As such, I think the work is promising, although I believe further analyses are needed before the data can be meaningfully interpreted, as explained below.

MAIN COMMENTS

Given the framing of the paper around "modeling saliency across individuals" I was surprised to see that all of the analyses are performed at a group level. If I have understood correctly, model performance is always evaluated in terms of the Normalised Scanpath Saliency score (NSS) between the model's predicted continuous saliency map and a human saliency map that is derived by (Step 1) aggregating many participants' measured fixations together in a single map, then (Step 2) applying a gaussian blur to make these discrete fixations into a continuous map. Three different analyses are reported, which differ in how participants are aggregated at Step 1: Table 1 reports model evaluations when all 8,235 participants are included; Table 2 reports model evaluations when participants are grouped into nine bins by age; and Supplementary Table 3 reports the same when participants are grouped into two bins by gender. The paper's main conclusions hinge on differences in model performances across the nine age bins. However, differences in the sample size and likely data quality between bins make it difficult to interpret model performance differences.

As a crucial first step, we need to see the baselines of Spatial Distribution Map (SDM) and Single Observer (SO) *within each age group* (i.e. what is the NSS of the bin-averaged spatial distribution map? And what is the average NSS when using each individual within that age group to predict the bin-averaged spatial distribution map?). These provide us with, respectively, the maximum possible performance of any model within that bin, and a proxy for the amount of inter-individual variability within that bin (which gives an estimate of how well a model would perform if it were as good as another representative human individual).

This is an excellent suggestion. We added SI Table 2 (below) to the supplementary material. Fixation maps constructed from gaze behavior of all participants within the same age bin, as well as the average single observer within each age bin were more predictive than the averaged single observer across all participants or fixation map across all observers. This implies that gaze behavior of e.g., 6-11 year olds is indeed more similar to the gaze behavior of other 6-11 year olds than of other age groups.

Model	M_{sample}	$M_{\text{(bins)}}$	6-11	12-17	18-23	24-29	30-35	36-41	42-47	48-53	54-59
Fixation map	0.71	0.834	1.02	0.862	0.837	0.731	0.729	0.729	0.769	0.816	1.013
Single observer	0.174	0.185	0.256	0.211	0.219	0.176	0.164	0.148	0.169	0.164	0.156
Central bias	0.001	-0.005	-0.036	0.002	-0.025	0.028	0.009	-0.011	-0.034	0.034	-0.013
Meaning map	0.382	0.367	0.338	0.343	0.409	0.414	0.382	0.365	0.361	0.361	0.33
RARE2012	0.194	0.182	0.163	0.147	0.185	0.226	0.197	0.188	0.185	0.216	0.129
SalGAN	0.42	0.402	0.395	0.334	0.471	0.459	0.406	0.408	0.426	0.408	0.315
DeepGazeII	0.405	0.392	0.396	0.392	0.44	0.435	0.401	0.361	0.378	0.407	0.32
SALICON	0.455	0.439	0.433	0.427	0.51	0.478	0.442	0.428	0.447	0.463	0.324
DVA	0.3	0.285	0.279	0.224	0.313	0.339	0.304	0.277	0.305	0.3	0.225
FES	0.052	0.043	0.051	0.014	0.039	0.083	0.064	0.042	0.029	0.059	0.007
QSS	0.338	0.328	0.365	0.294	0.363	0.364	0.338	0.304	0.332	0.337	0.259
SSR	0.265	0.252	0.257	0.206	0.297	0.278	0.283	0.257	0.246	0.28	0.163
CVS	-0.085	-0.087	-0.067	-0.076	-0.102	-0.077	-0.074	-0.091	-0.11	-0.079	-0.111
IMSIG	0.333	0.317	0.33	0.282	0.353	0.361	0.347	0.306	0.323	0.334	0.221
LDS	0.031	0.027	0.045	0.048	0.027	0.048	0.028	0.014	-0.003	0.037	-0.004
ICF	0.26	0.251	0.252	0.275	0.268	0.288	0.258	0.21	0.233	0.282	0.192
GBVS	0.166	0.156	0.134	0.171	0.151	0.207	0.157	0.142	0.13	0.2	0.113
CAS	0.161	0.154	0.177	0.171	0.158	0.18	0.161	0.135	0.134	0.187	0.086
SUN	0.167	0.165	0.183	0.147	0.161	0.183	0.176	0.141	0.147	0.206	0.144
DeepGazeI	0.334	0.319	0.322	0.311	0.363	0.365	0.328	0.294	0.311	0.346	0.232
AIM	0.26	0.259	0.283	0.264	0.269	0.276	0.259	0.21	0.237	0.297	0.234
SAM	0.258	0.255	0.254	0.337	0.288	0.27	0.215	0.225	0.236	0.288	0.182
DeepGazeII	0.4	0.382	0.373	0.341	0.42	0.446	0.396	0.37	0.383	0.393	0.317
IKN	0.202	0.188	0.174	0.164	0.194	0.247	0.202	0.18	0.171	0.223	0.135
BMS	0.186	0.175	0.216	0.182	0.202	0.201	0.192	0.159	0.155	0.194	0.074

Table 2

Absolute NSS values per baseline and model across age bins. M_{sample} is the average performance across all participants analyzed here, M_{bins} is the average NSS across age bin averages per baseline/model.

Plotted, these data look as follows:

or as follows if the fixation map is included:

We refer to this finding in the manuscript as follows:

“Deviations across all age bins and models are given in Table 2 and depicted graphically in Figure 2; spatial distribution maps of fixation locations are given per age bin in SI Figure 2; absolute NSS per model and baselines are given in SI Table 2.”

Absolute NSS for female and male participants per model can also be inferred from SI Table 5.

Comparing the central bias to gaze behavior of each bin/gender suggests that a difference in centrality/spread is a fundamental driver of biases across demographic groups. We adjusted the manuscript and supplementary material in several parts.

In the results on gender effects:

“The average difference in predictions across gender for all models (NSS=0.017) closely corresponded to the difference for the central bias (NSS=0.018, SI Table 5).”

In the results on age effects:

“Again, the central bias showed a largely similar tendency to the overall model biases, in line with findings reported earlier on a smaller selection of models and age groups (Açık et al., 2010; Krishna & Aizawa, 2017).”

And in the discussion:

“Overall, a large portion of the biases in predictions across demographic groups followed relative differences in prediction of the central bias baseline, in line with previous work with smaller samples (Açık et al., 2010; Krishna & Aizawa, 2017; Krishna et al., 2018; Rider et al., 2018).”

The peak performance of the best model within any age group appears to be 0.081 (QSS, 24-29yo, Table 2), which is substantially less than the performance of the best model when assessed using all of the data (0.226, Table 1). This makes sense, because the age-binned spatial distribution maps are likely noisier since they are created using far smaller sample sizes (max bin N = 338) than the all-participant data (N = 8,235). Most importantly, the noisiness of the data almost certainly differs between age bins, both because sample sizes vary (from N = 36 to N = 338) and because participants of different ages likely give differently noisy data (e.g. young children may look for shorter durations,

or have greater head motion). These are inevitable imperfections in any such large dataset gathered "in the wild" and are not necessarily a problem. *However* they mean that to meaningfully interpret the data, the baseline/ceiling values (for SDM and SO) should be shown separately for each age group, and ideally model performances should be expressed as a proportion of the explainable "signal" available in the data (e.g. model NSS divided by SO NSS, within each bin).

We are sorry for having been unclear here. Table 2 indexes not *absolute* NSS, but *relative* deviations to the average NSS across all age bins per model. This way, imbalances in sample size should also be accounted for in their effect on the mean. Therefore, the value for QSS didn't represent an absolute NSS, but the strongest positive deviation – in other words, QSS was most strongly biased towards this age group.

Following the reviewer's recommendation, we now also ran the single observer model against the map constructed by the fixations of all other observers of the same demographic bin, i.e., per age group and once for females and once for male participants (see previous point). This can also be seen as evidence that the number of participants in all age bins has been sufficient to provide a stable fixation map. Absolute NSS per model and age bin, as well as the absolute NSS of the fixation map per age bin (the upper bound) are given in SI Table 2. We further added a subplot to Figure 2, in which we expressed the NSS scaled in % between the maximally achievable NSS (fixation map) and the single observer NSS). Biases across age are even stronger here. We also added SI Table 3 with significance tests across age bins for the data visualized in the bottom subplot of Figure 2, little surprisingly, comparisons are quite clear statistically for most (see below). We refer to these findings at several places in the results and discussion section.

Figure 2

Table 3

Significance tests for percentage explained NSS of the explainable NSS given in Figure 2 bottom across age bins.

Comparison	Mean Difference	t	p (Bonferroni)	95% CI		
				95% CI (Lower)	95% CI (Upper)	
6-11	12-17	-3.889	-2.548	0.424	-8.854	1.077
	18-23	-8.191	-5.367	< .001	-13.157	-3.225
	24-29	-19.075	-12.499	< .001	-24.041	-14.11
	30-35	-16.039	-10.509	< .001	-21.004	-11.073
	36-41	-14.151	-9.272	< .001	-19.117	-9.185
	42-47	-11.345	-7.434	< .001	-16.311	-6.379
	48-53	-16.375	-10.73	< .001	-21.341	-11.409
12-17	54-59	-3.811	-2.497	0.487	-8.777	1.154
	18-23	-4.302	-2.819	0.195	-9.268	0.663
	24-29	-15.187	-9.951	< .001	-20.153	-10.221
	30-35	-12.15	-7.961	< .001	-17.116	-7.184
	36-41	-10.262	-6.724	< .001	-15.228	-5.296
	42-47	-7.457	-4.886	< .001	-12.423	-2.491
	48-53	-12.487	-8.182	< .001	-17.452	-7.521
18-23	54-59	0.077	0.051	1	-4.889	5.043
	24-29	-10.885	-7.132	< .001	-15.85	-5.919
	30-35	-7.848	-5.142	< .001	-12.813	-2.882
	36-41	-5.96	-3.905	0.005	-10.926	-0.994
	42-47	-3.154	-2.067	1	-8.12	1.811
	48-53	-8.184	-5.363	< .001	-13.15	-3.218
	54-59	4.379	2.87	0.168	-0.586	9.345
24-29	30-35	3.037	1.99	1	-1.929	8.003
	36-41	4.925	3.227	0.055	-0.041	9.891
	42-47	7.73	5.065	< .001	2.764	12.696
	48-53	2.7	1.769	1	-2.265	7.666
	54-59	15.264	10.002	< .001	10.298	20.23
30-35	36-41	1.888	1.237	1	-3.078	6.854
	42-47	4.693	3.075	0.089	-0.273	9.659
	48-53	-0.337	-0.221	1	-5.302	4.629
	54-59	12.227	8.012	< .001	7.261	17.193
36-41	42-47	2.805	1.838	1	-2.16	7.771
	48-53	-2.224	-1.458	1	-7.19	2.741
	54-59	10.339	6.775	< .001	5.373	15.305
42-47	48-53	-5.03	-3.296	0.044	-9.996	-0.064
	54-59	7.534	4.937	< .001	2.568	12.5
48-53	54-59	12.564	8.233	< .001	7.598	17.529

It would also be good to mention why a group analysis is necessary at all - if interested in predicting gaze in individuals, why not measure each model's NSS against each individual's blurred fixation map (and using the group-average SDM to estimate the ceiling of possible model performance)? Presumably because the sparseness of fixations and high variation across individuals renders such an analysis very noisy?

We concur that this is an interesting question. Indeed, just ten seconds (or even less, 18 fixations) of free viewing unfortunately results in relatively sparse maps per individual – and therefore very noisy NSS scores.

MINOR COMMENTS

- It would be good to provide a bit more detail in the Introduction on some of the most widely-used saliency benchmark datasets: How many images and participants? Does each participant look at each image, or do the data for different images come from different participants? What is the age range of participants? Are models standardly evaluated against the participant-average fixation map for each image, or against each individual participant for each image? This information would help the reader appreciate the differences between common benchmarks and the current dataset.

We thank the reviewer for this sensible suggestion. There is some variation across data sets regarding those factors (some of which might underlie different benchmarking rank orders, see the newly added part on generalization across fixations). Generally, however, the large and highly influential CAT2000 dataset can be seen as *pars pro toto* and gives a good summary of demographics and general approach. Note that many datasets do not give average ages, but age ranges – which can be assumed to be heavily skewed. Hereby, models are evaluated against participant average maps per image, although this isn't always specified.

“Researchers in turn benchmark their models on empirical data which usually contain fixations of a limited number of adult individual participants (e.g., Coutrot & Guyader, 2014; Judd et al., 2012; Judd et al., 2009) who view a very wide range of static images without instruction (see Kümmerer, Bylinskii, et al., 2022, for an overview). For instance, the influential and vast CAT2000 dataset (Borji & Itti, 2015) contains free viewing data of 120 participants (80 female, 40 male) aged 18-27 years with each of 4,000 images being viewed for 5 s by 24 participants.”

- Briefly describe the data-collection context early in the paper (i.e. around Figure 1 in the Introduction). The fact that it was collected at a museum via an interactive exhibit helps the reader understand why just a single image was used, and how such a large sample size was obtained.

We agree that this helps to get the nature of the data conveyed early on and set expectations right. We think that the figure caption is ideal to do so and write:

“Usable gaze data were obtained from $n=2,607$ participants using an eyetracker and monitor that was part of an installation at a Science museum (see SI Figure 1 for pictures of the setup).”

We address the single image more explicitly in the discussion now:

“However, many models already incorporate a central bias and the image at hand features many spread-out objects.”

And

“Replications of presently reported findings across multiple images would be desirable, for instance using other large scale data collections such as described here.”

To make the special character of the data collection and setup more transparent, we now also included an additional figure in the supplementary material, featuring pictures of the setup (see below).

SI Figure 1. Setup at the science museum. Upper left and upper center: Metal box containing the eye tracker and monitor. Upper right: view inside the metal box, 'horns' are loudspeakers used after free viewing to ask for data donation. Bottom left: starting screen. Bottom center and right: participant taking part in the study. Participants sat on a small chair..

- Table 1: Provide more information in the table caption. e.g. briefly explain "NSS", what "% improvement" is relative to, and what "spatial distribution map" refers to.

We now added a clarifying caption as follows:

“Model performance (NSS) for baselines and models are given as rows. Negative numbers denote worse than chance performance. The improvement column denotes relative performance between central bias (0%) and fixation map (100%) in % NSS.”

- p11. It's stated that participants were excluded from demographic analyses if they didn't change the default settings (birth year = 2000, gender = nonbinary), but it's not clear whether participants were excluded if *either* of these was unchanged (which would systematically exclude all nonbinary people from analyses), or only if *both* of them were unchanged (which would exclude the much smaller subset of nonbinary people born in 2000).

Indeed, we excluded participants if either non-binary OR 2000 as date-of-birth were entered, instead of AND. Given the large size of the dataset, we would prefer to remain as careful as possible here. Even though it would be nice to gain a few participants with correct information, we are worried about still including less reliable datasets. Of course, our data would be interesting with non-binary gender as additional group, but here we would only gain 91 participants, distributed across the 6-59 age range. We label non-binary as such, but some participants might also have ended up in this group because they did not want to indicate their gender. We now state this in the paper as follows:

“N = 91 participants indicated non-binary gender across the 6-59 age range, but given that there was no option for 'prefer not to say', these data were not analyzed.”

- The proportion of participants who reported demographics is very low (1,412 out of 8,235 = 17%). This is understandable - participant commitment in an interactive museum exhibit is likely to be low, people may be reluctant to provide personal information, and the gaze-contingent method of reporting demographics (though potentially engaging) may be unweildy. However, it would be good to acknowledge this limitation in the discussion, and perhaps comment on whether certain age groups may have been more likely to provide demographic information than others, leading to

systematic over-representation of some age groups in the no-information majority of the dataset.

Even though we now report new data (with a much higher ratio of inclusion, 1,600 out of 2,607 participants), certain age groups are almost certainly overrepresented. Relative to the demographics of the visitors to the museum (which is targeted at children and visited by many school classes), our average age is relatively high. This is likely due to the humanities part of the museum being targeted at a slightly older population and it takes some patience to complete the experiment. Age groups can also induce further sampling biases. For instance, children might only be able to participate if they can be sufficiently patient or older participants need to have sufficient vision. We fully agree that it is sensible to acknowledge this explicitly and now write:

“Naturally, the sample put forward here, while vast, is in turn affected by sampling biases. For example, children need to be willing to wait and focus, older adults need to have sufficient vision to participate. Which further differences between people, beyond gender and age, are relevant when it comes to saliency maps remains to be determined.”

- To the extent that findings are inconsistent with previous reports in the literature (e.g. overall poor performance of most models relative to the centre prior baseline), how much might this be due to the unusual nature of the single probe image? Most work on developing saliency models has considered "natural images" (i.e. human-taken photographs), which have a quite different spatial structure and distribution of "meaningful elements" than this digital collage image. It could even be argued that the use of such "unnatural" images is a strength, in that they can showcase aspects of gaze behaviour that are usually neglected in saliency research. However, it would be good to consider this point in the Discussion.

Indeed, some findings will not generalize to all images necessarily. Using a particularly feature rich image, like the one at hand, should provide a stronger test to differences in model performance than feature-sparse images (as there are more factors determining gaze behavior). In a certain sense, this single-image approach could reveal some problems in saliency modeling (as present in the newly added across-fixations effects) that could easily have been overlooked otherwise. We discuss this more explicitly throughout the discussion in particular now. Also note that the model performances overall have improved quite a bit using the new data – unlike the model biases.

Reviewer #2 (Remarks to the Author):

This paper evaluates a range of salience models against the fixations of more than 8,000 observers, freely viewing a single image in a museum setting. Additionally, it tests whether model performance varies as a function of gender and age, using a subset of more than 1,400 observers who provided corresponding data.

In general, the performance of salience models was poor. Fewer than half of the tested models outperformed a simple centre prior. Notably, three models - including GBVS - outperformed a 'meaning map'. The best performing model (DeepGazellE) was far worse than the average performance for using a single observer's data to predict the fixations of all others.

There was a non-significant trend for better model performance for male vs female observers and significant variation across age bins. Model performance peaked for young adults and was particularly bad for children below the age of 12. The authors conclude that a narrow focus on college student populations runs the risk of myopic models and overfitting.

Strengths:

The presented dataset, to the best of my knowledge, is the largest of its kind. Processed fixation data is already available via OSF and the authors seem to plan to release the raw data as well, which would render this a highly valuable resource for the field.

Although previous studies compared free-viewing data of children and adults (including the fit of low-level salience; e.g. Acik et al., 2010 Front Psychol), the current sample allows to quantify the fit to salience models over an unprecedented range with reasonable granularity. It is particularly interesting that model performance peaks around the age range typically sampled in previous studies and drops off at either side of this. The authors' call for awareness of sampling biases and their drawbacks is well justified.

We included this most relevant reference in addition to other previously omitted work.

Given the large sample size and high power, the absence of significant gender differences is meaningful and interesting in its own right.

The observation that GBVS outperformed meaning maps is a relevant contradiction to previously published results.

Problems to be addressed:

Quality of fixation data

I'm unsure about the quality of fixation data. A median of 11 fixations appears extremely low for 10s viewing time – less than half of what would typically be expected. I suspect this is related to the fixation definition requiring a minimum gap of 60ms between successive fixations. Typical saccade durations are 20-40ms, so this seems a highly surprising parameter choice which may well lead to a loss of 50% of fixations. Without access to the raw data it is hard to judge what is going on, but a look at the processed fixation data reveals systematic gaps of hundreds of ms between successive fixations. Given all other analyses hinge on this, this point strikes me as crucial. Please double-check the procedure and provide example X,Y traces and their segmentation into events for a couple of observers and trials. If my suspicion is warranted, the raw data should be re-analysed using appropriate parameters, which may or may not change the pattern of results (e.g. if children make shorter saccades a higher fraction of fixations may have been dropped because of shorter inter-fixation gaps).

We would like to sincerely thank the reviewer for spotting this flaw (see general point above). Indeed, data were smoothed, which is why we decided to discard the data altogether and re-assess usable data for $n > 2,600$ participants, which we now report. The 18 fixations observed and analyzed within about 7s are plausible given the viewing duration (median number of fixations across 10s of freeviewing = 25.0; median fixation duration = 200.5 ms). Please see raw gaze traces and event detection outcomes in the next reply.

The surprisingly small number of fixations may also be down to excessive data loss. The authors claim that ‘data loss cannot be assessed with the current setup’ (p. 11), shortly after explaining that ‘no periods of more than 5s of lost gaze position’ (p.10) served as an exclusion criterion. This seems contradictory and is in need of explanation. The proportion of samples with lost gaze position should be quantified and reported.

We could now investigate data loss directly. Data exclusion remained otherwise unchanged. In sum, data quality measures loss and precision do not follow a pattern of results that would explain the findings observed here for gender and age. We report data loss as follows and added Figure 4 to the supplementary material and also report data loss in the methods section of the main article:

“Eye tracking data quality can be assessed by precision, accuracy, and data loss (Dunn et al., 2023; Holmqvist et al., 2022). While accuracy cannot be assessed with the current setup, precision, calculated as in Hooge et al. (2018), was $Mdn = 0.68^\circ$ ($SD = 0.28^\circ$), loss was $M = 0.8\%$ ($SD = 2.4\%$). These are reasonable values given the special nature of the setup (see SI Data quality for more information and SI Figure 5). Neither precision nor loss have visibly driven results across demographic groups.”

SI Figure 4. Precision across age bins and gender in RMS (upper) and data loss across age bins and gender (middle row). Scatter of precision against data loss across age bins (bottom).

And in the supplementary information:

“Median precision was 0.68° with a standard deviation of 0.28° . Data loss was $M = 0.8\%$, $SD = 2.4\%$. Female participants had an average data loss of 0.78% and male participants had an average data loss of 0.79% . SI Figure 4 depicts data quality (precision in RMS and percentage data loss) across age bins and indicated gender. From the overall pattern of data quality indicators, data quality seems unlikely to have driven the main biases reported in the manuscript.”

The authors further report precision estimates based on inter-sample distance. This rests on the assumption that successive samples are independent estimates of position. Having worked with the 4c, this seems unlikely. Tobii doesn't provide information about their proprietary filtering algorithms, but the position estimates appear visibly 'sticky', as in a moving average or Kalman filter. This of course would render inter-sample distance an inappropriate proxy to precision. As long as there's no evidence of successive samples being truly independent measures, this precision proxy should be avoided.

Indeed, we fell for this 'sticky' signal – and its drawbacks. The newly reported data do not suffer from this drawback (see the figure with event detection and raw gaze signal traces below). Again, we would like to thank the reviewer for spotting this major mistake. We added raw gaze traces in combination with the resulting detected events of six random participants below for transparency (additional timeseries figures can be found on OSF). The new data do not seem to suffer from smoothing that is inherent to the interaction signal output by the Tobii 4C.

Finally, the authors used a single point calibration procedure and did not collect any validation data. At the same time the screen (and image? See below) covered a vast visual angle. I'm not overly

worried regarding this point, because of personal experience with the 4c and the reasonable looks of the heatmaps. But it should be feasible to collect proper calibration and validation data for the full screen post-hoc. Data from just a few participants (ideally children and adults) would render the authors case for data quality much more compelling.

Unfortunately, we could not directly fulfill this request as we could not access this data due to technical implementation of the set-up in the museum. However, we did run tests in which the authors fixated edge points on the screen to test for the calibration accuracy, qualitatively, these data matched fixated locations. Note that we now used a 5 point calibration instead of a one point calibration procedure to ensure higher data quality.

“Participants were required to look at a central circle that gradually filled to start the experiment and perform a five-point calibration of the eye-tracker.”

Single image

The use of a single image means we don't know how well the results generalize. This is compounded by the peculiar nature of the collage image, which renders it different from a single natural scene. The heatmaps appear clearly driven by the collage nature of the stimulus. This should be discussed.

We agree (with R1 and 2) that this point deserves discussion. We discuss this multiple times throughout the manuscript:

We address the single image more explicitly in the discussion now, e.g.:

“However, many models already incorporate a central bias and the image at hand features many spread-out objects.

and

“Replications of here reported findings across multiple images would be desirable, for instance using other large scale data collections such as described here.”

*Unequal sampling *

The number of data points per bin is extremely unequal between age groups and particularly low for children under 12. Could this be a confound? To be sure, the authors could randomly subsample an equal number of observers from the older bins and in each iteration calculate the NSS for the subsample. If the observed NSS for children is well outside the distribution of values observed for random adult subsamples of the same size, this confound can be excluded.

We thank the reviewer for this important point. We looked at several options, including the suggested bootstrapping procedure. For simplicity, we opted to report the extracted central tendency per age groups, which similarly alleviates the concern.

To address this issue, we now also report model performances, as well as the maximum achievable NSS per age group (SI Table 2; see above response to R1). In fact, when giving NSS as a proportion of the overall max achievable NSS, age biases are even stronger, not weaker overall (now added to Figure 2, color-coded).

A further indication that the reported results are not just the consequence of unequal sample sizes per bin is that the average of single observer models per age bin reached a better NSS than the single observer model across all participants. In other words, more demographically

similar participants yielded more similar fixation patterns. The pattern hereby did not follow the bias we observed in model performance across age.

We now also added baselines per age group to Table 2 to the main manuscript and included overall NSS per model and baseline per age bin in the supplement.

We further added to the manuscript:

“Note that, due to smaller group sizes, the extreme age bins 6-11 and 54-59 warrant more caution before interpretation than the other age bins.”

Unknown demographics for model comparisons

The comparison of model performances rests on data with unclear demographics. Therefore, it remains open which role sample specifics play for these general observations. For example, does GBVS outperform meaning maps in all age brackets? Supplementary analyses zooming in on the data with known demographics could shed light on this. A good way of showing this would be to provide a version of figure two colour coding all models (plus baselines) without obscuring them by the median line. Relatedly, for the youngest bin the median line seems to fall below the majority of models (bit hard to tell, but I count at least 12 lines ending above the median, which doesn't seem to add up?).

We are sorry for having been confusing here. The black line corresponded to the right y axis and gave median % deviations. To avoid this confusion, we removed % illustrations and now depict mean and confidence interval of the mean in the updated Figure 2. We agree that this information is useful to the readers and we added the new SI Table 2 with the absolute NSS values per model and age bin for transparency, alongside all baselines. We further added the baselines to Table 2 of the main manuscript. Figure 2 bottom now features a color-coded plot.

We plotted this information color coded below (see also reply to R1):

More details on methods, please

We thank the reviewer for the ideas on sensible additional details, which we added to the methods section/supplement (see below for further details per individual point):

The methods section is very sparse and should convey more detail. For example: Did image size correspond to screen size?

“Participants were presented with a full-screen collage image”

What kernel was used for smoothing fixation maps?

Was a kernel used for NSS fixation sampling, or did a fixation correspond to a single pixel?

“1) A map of all actual fixation locations served as the upper bound for the performance of any model (comparison between the binary array of fixation locations and its smoothed counterpart). This fixation map was constructed from fixation locations by applying a Gaussian filter (SD =1 degree of visual angle) to the fixation map, effectively making it continuous (Bylinskii et al., 2018; Le Meur & Baccino, 2013) - in other words, discrete fixation locations were blurred over with this kernel. .“

For computing NSS scores, each fixation corresponded to a single pixel. We added:

“NSS was extracted per model by first z-standardizing the respective saliency map, and overlaying it with the binary map of discrete fixation locations. For each fixated pixel, the z-score of the corresponding pixel was taken from the saliency map and a grandmean was calculated over those values.”

What was the physical setup?

We added pictures of the setup to make the nature of the setup more transparent in the supplement. We also refer to this figure in the main article.

SI Figure 1. Setup at the science museum. Upper left and upper center: Metal box containing the eye tracker and monitor. Upper right: view inside the metal box, 'horns' are loudspeakers used after free viewing to ask for data donation. Bottom left: starting screen. Bottom center and right: participant taking part in the study. Participants sat on a small chair.

Did observers stand or sit?

Participants were free to either stand or sit, however, the setup was not height adjustable. A chair (see above) could be used to either sit or stand on. See below for adjustments in text.

How were size differences in children accounted for?

Size differences were not accounted for directly. Practically, however, we can assume that shorter participants (mostly children) were generally standing more than longer participants (mostly adults) that were more likely to stand. Very short children could stand on the chair (see the newly added SI Figure 1 for pictures of the setup). See below for adjustments in text.

What were the instructions given via the speakers?

No instructions were given prior free viewing. The only text displayed upfront was “What are you looking at?” in English or Dutch. After participants selected their preferred language, a calibration was performed. Speakers were used to support a video in which the last author asked for data donations after free viewing (video including subtitles).

“A metal box around screen and tracker shielded the field of view from other visual stimulation. Participants could either stand, sit on a chair, or stand on a chair to be able to see the monitor and participate. Other than that, the setup was not height adjustable. Auditory information was given exclusively after free viewing via two loudspeakers positioned close to the participants’ ears. See SI Figure 1 for pictures of the setup..”

Also – are you sure about a sampling rate of 60 Hz – the default for the 4c is 90 Hz.

Indeed the tobii sampled at 90Hz. The algorithm to extract and save this data, however, was running at 60Hz (asynchronous sampling, i.e., the last available sample was used).

“Gaze was logged at 60 Hz using a Tobii Eye Tracker 4C (asynchronous tracking)”

Previous developmental studies

Please relate your findings to those of previous developmental studies (you cite some) and demarcate the advance of the present study compared to them.

We agree that the findings needed to be embedded better and have now referenced to relevant literature in this regard in both introduction and discussion. What makes the present work unique is the vast sample that allows for much more fine grained insights into age bins than existing work (such as Açık et al., 2010). Furthermore, the special character of the data collection might lead to overall less biased samples than university contexts – albeit differently biased. Lastly, the newly added across fixations analyses are only possible with a vast number of fixations per number of fixation (i.e., per first, second fixation and so forth), which is arguably impossible unless sample size is in the hundreds due to sparsity.

“This considerably expands existing works on generalizability across age (Açık et al., 2010; Krishna & Aizawa, 2017; Krishna et al., 2018; Rider et al., 2018), which feature much smaller samples and more distinct age groups..”

“Knowledge about developmental differences in sampling behavior and saliency computations (Açık et al., 2010; Franchak et al., 2016; Gottlob & Madden, 1999; Mitchell & Neville, 2004; Rider et al., 2018) could be incorporated here, as well as findings into differences between earlier and later fixations (e.g., Ossandón et al., 2014; Pannasch et al., 2008).”

“Notably, variation in model performance was relatively large for under-aged (< 18 year-old) participants compared to other participants. Again, the central bias showed a largely similar tendency to the overall model biases, in line with findings reported earlier on a smaller selection of models and age groups (Açık et al., 2010; Krishna & Aizawa, 2017).”

Methods contain a doubled sentence on highlighting of three most fixated sub-images.
We thank R2 for spotting this mistake, fixed.

Reviewer #3 (Remarks to the Author):

The manuscript by Strauch et al. investigates generalizations in saliency predictions across a very large number of individuals. Overall, model performance is well, however, breaks down for younger and older age groups than the most often studied group containing university students. These results are important and interesting for a wide audience.

The manuscript is clearly and succinctly written. The literature is adequately cited and the Methods seem sound. The team of author clearly represents high expertise. I have no major critique.

We are happy to read the positive assessment of R3 and hope that the substantially revised manuscript finds similar acclaim.

I wonder if the term "centre prior" could be misleading for some readers, since it might suggest Bayesian terminology, which is, however, not the case here. Therefore, I suggest to use "centre bias" instead.

We agree with R3 and now changed it to central bias consistently.

The language should be checked with respect to use of English vs. American (e.g., no consistent use of "modeling" vs. "modelling").

We adjusted to American English and hope to have spotted the inconsistencies.

17th Aug 23

Dear Dr Strauch,

Thank you for your patience during the peer-review process. Your manuscript titled "None to rule them all? Systematic shortcomings in modeling saliency across individuals and fixations" has now been seen by 2 of the original reviewers, and I include their comments at the end of this message. They are both impressed by the quality and the extent of your previous revisions. We are interested in the possibility of publishing your study in *Communications Psychology*, but would like to consider your responses to these concerns and assess a revised manuscript before we make a final decision on publication. Reviewer 2 has some concerns that would like to see addressed in a final revision. We highlight in particular that there would be the need of a missing control analysis, or alternatively a discussion of the implications of this missing control in the discussion. Moreover we would like to see a more nuanced discussion of the caveats, in the section titled "Limitations" that is part of the Discussion section.

Editorially, we ask that you include an additional analysis in the Supplement in which you display the data for the participants who did not de-select the option "non-binary". Although the data cannot be interpreted because the exclusion procedure does not differentiate between participants who indicated that they are non-binary from those who did not change the preset option for other reasons, because the number of exclusions is considerable, they should be displayed. We would also like to see a discussion around this limitation.

Please submit a revised manuscript along with a point-by-point response to Reviewer 2 working with the template and checklist (for which I include links below) to ensure that your manuscript fully complies with our reporting and formatting standards. Please highlight all changes in the manuscript text file.

Please use the following link to submit your revised manuscript, point-by-point response to the referees' comments (which should be in a separate document to any cover letter) and the completed checklist:

[link redacted]

We would appreciate it if you could keep us informed about an estimated timescale for

resubmission, to facilitate our planning.

Please do not hesitate to contact me if you have any questions or would like to discuss these revisions further. We look forward to seeing the revised manuscript and thank you for the opportunity to review your work.

Best regards,

Eva R. Pool

Eva R. Pool, PhD
Editorial Board Member
Communications Psychology
orcid.org/0000-0001-5929-1007

EDITORIAL POLICIES AND FORMATTING

Editorial Policy: [Policy requirements](https://www.nature.com/documents/nr-editorial-policy-checklist.pdf) (Download the link to your computer as a PDF.)

Furthermore, please align your manuscript with our format requirements, which are summarized on the following checklist:

[Communications Psychology formatting checklist](https://www.nature.com/documents/commspsychol-style-formatting-checklist-article-rr.pdf)

and also in our style and formatting guide [Communications Psychology formatting guide](https://www.nature.com/documents/commspsychol-style-formatting-guide-accept.pdf) .

*** TRANSPARENT PEER REVIEW:** Communications Psychology uses a transparent peer review system. This means that we publish the editorial decision letters including Reviewers' comments to the authors and the author rebuttal letters online as a supplementary peer review file. However, on author request, confidential information and data can be removed from the published reviewer reports and rebuttal letters prior to publication. If your manuscript has been previously reviewed at another journal, those Reviewers' comments would not form part of the published peer review file.

*** CODE AVAILABILITY:** All Communications Psychology manuscripts must include a section titled "Code Availability" at the end of the methods section. In the event of publication, we require that the custom analysis code supporting your conclusions is made available in a publicly accessible repository; at publication, we ask you to choose a repository that provides a DOI for the code; the link to the repository and the DOI will need to be included in the Code Availability statement.

Publication as Supplementary Information will not suffice. We ask you to prepare code at this stage, to avoid delays later on in the process.

*** DATA AVAILABILITY:**

All Communications Psychology manuscripts must include a section titled "Data Availability" at the end of the Methods section or main text (if no Methods). More information on this policy, is available at <http://www.nature.com/authors/policies/data/data-availability-statements-data-citations.pdf>.

At a minimum the Data availability statement must explain how the data can be obtained and whether there are any restrictions on data sharing. Communications Psychology strongly endorses open sharing of data. If you do make your data openly available, please include in the statement:

We recommend submitting the data to discipline-specific, community-recognized repositories, where possible and a list of recommended repositories is provided at <http://www.nature.com/sdata/policies/repositories>.

If a community resource is unavailable, data can be submitted to generalist repositories such as [figshare](https://figshare.com/) or [Dryad Digital Repository](http://datadryad.org/). Please provide a unique identifier for the data (for example a DOI or a permanent URL) in the data availability statement, if possible. If the repository does not provide identifiers, we encourage authors to supply the search terms that will return the data. For data that have been obtained from publicly available sources, please provide a URL and the specific data product name in the data availability statement. Data with a DOI should be further cited in the methods reference section.

REVIEWERS' COMMENTS:

Reviewer #1 (Remarks to the Author):

The authors have done an absolutely stellar job with this revision. Having noticed an error in the data preprocessing, they have completely redone all analyses with a new sample of over 2,500 participants. The new data is cleaner and has a higher proportion of reported demographics, and the authors have also implemented several new analyses and added sections to the manuscript in response to the previous round of reviews. It is heartening to see such a large project replicated before it even goes to print!

The catalogue of saliency models generally do somewhat better on this unsmoothed dataset than on the previous one, which is reassuring. But the overall message of the work remains the same - state of the art saliency models may not perform as impressively as originally reported, when tested on a much more diverse range of participants, and on a non-standard image. There appear to be reliable age and gender differences in viewing behaviour that are not accounted for by such models.

I also appreciate the effort the authors have gone to to evaluate the performance of single-observers of each age bin (SI Table 2) and to express NSS in terms of the proportion of potentially explainable variance in each age group (Figure 2 subplot). These additions help assuage my (and hopefully other readers') concerns that age group differences could be artefacts due to different sample sizes.

Congratulations on a huge but excellent overhaul of the paper!

Reviewer #2 (Remarks to the Author):

I commend the authors for their corrections to the previous version and efforts to collect new data. In my view, the replies to my previous remarks render the reported age and gender effects much more convincing.

However, I'm concerned about the new discussion of model performance for early vs later fixations. While the effects of age converge across absolute model performance and performance normalised to the estimated maximum, this is not the case for early and late fixations. There's a clear divergence between absolute model performance (higher NSS for `_early_` fixations) and performance relative to the estimated maximum performance (higher relative NSS for later fixations). In other words, model predictions are **better** for early fixations, but closer to a (much) lower noise ceiling for later fixations.

This calls for a careful and nuanced discussion, which currently is lacking. The authors seem to largely ignore absolute model performance and almost exclusively focus on performance normalised to the estimated maximum. Especially the divergence between the absolute and normalised measures leads to questions relevant to the interpretation of findings. For instance: How sensitive is maximum performance to the choice of smoothing kernel size? How dependent is normalised model performance on cut-off values when thresholding salience maps? Could salience models using the same set of features and weights improve normalised performance for the first fixations if they used more greedy thresholding on their salience maps? How much of the higher entropy for later fixation maps is due to replicable inter-individual variance and how much to higher levels of randomness (i.e. in a repeat experiment would fixation 18 of subject A have a lower likelihood to be the same as fixation 1)?

It would also be good to show model performance for individual fixations rather than cumulated across the first n fixations only (this is mentioned in the text for a comparison of fixation 9 and the first two, but I can't seem to find it in the figures).

Finally, some of the updated text appears to contain mistakes (e.g. legend of Figure 2 'scaled NSS scaled between...'), so a careful proofread would be in order. Figure S6 and the following page seem

corrupted in the PDF

Point by point response

We were happy to read that both reviewers assessed the revised manuscript so positively. In the following, we reply to the outstanding points one by one.

Green: Our answers

Blue: Edits in manuscript

Reviewer comments:

Reviewer #1 (Remarks to the Author):

The authors have done an absolutely stellar job with this revision. Having noticed an error in the data preprocessing, they have completely redone all analyses with a new sample of over 2,500 participants. The new data is cleaner and has a higher proportion of reported demographics, and the authors have also implemented several new analyses and added sections to the manuscript in response to the previous round of reviews. It is heartening to see such a large project replicated before it even goes to print!

The catalogue of saliency models generally do somewhat better on this unsmoothed dataset than on the previous one, which is reassuring. But the overall message of the work remains the same - state of the art saliency models may not perform as impressively as originally reported, when tested on a much more diverse range of participants, and on a non-standard image. There appear to be reliable age and gender differences in viewing behaviour that are not accounted for by such models.

I also appreciate the effort the authors have gone to to evaluate the performance of single-observers of each age bin (SI Table 2) and to express NSS in terms of the proportion of potentially explainable variance in each age group (Figure 2 subplot). These additions help assuage my (and hopefully other readers') concerns that age group differences could be artefacts due to different sample sizes.

Congratulations on a huge but excellent overhaul of the paper!

We are happy to read this very positive assessment and thank the reviewer for the review on our manuscript.

Reviewer #2 (Remarks to the Author):

I commend the authors for their corrections to the previous version and efforts to collect new data. In my view, the replies to my previous remarks render the reported age and gender effects much more convincing.

However, I'm concerned about the new discussion of model performance for early vs later fixations. While the effects of age converge across absolute model performance and performance normalised to the estimated maximum, this is not the case for early and late fixations.

We are happy to read that the revised version has made the original manuscript stronger. We answer to the remaining points in the following.

There's a clear divergence between absolute model performance (higher NSS for `_early_` fixations) and performance relative to the estimated maximum performance (higher relative NSS for later fixations). In other words, model predictions are **better** for early fixations, but closer to a (much) lower noise ceiling for later fixations. This calls for a careful and nuanced discussion, which currently is lacking. The authors seem to largely ignore absolute model performance and almost exclusively focus on performance normalised to the estimated maximum. Especially the divergence between the absolute and normalised measures leads to questions relevant to the interpretation of findings.

We agree with the reviewer that the divergence between absolute and relative performances must be discussed here directly - and we agree that this improves the manuscript. In line with the new visualizations added here, and the useful remarks by the reviewer, we now stick to a more nuanced and careful tone in the discussion of these findings and the respective conclusions. We therefore rewrote the respective results and discussion sections, but also sentences in the abstract, e.g.:

“Model performance was characterized by substantial variation in NSS, but revealed that fixations are predicted differently well over time. Here, the fixation map, reflecting an upper bound, showed much higher NSS scores early than later on, likely because fixations were much more focal for early viewing (see Figure 3 bottom right). Models predicted the subset of early fixations generally better than when consecutive fixations were added, but only in absolute NSS scores (Figure 3 top left). When scaled to the maximum achievable NSS, many models improved relative to this maximum as consecutive fixations were added, given that the NSS scores of the fixation map and central bias also decreased (Figure 3 top right). Relative performances across models differed as a function of the cumulative fixations made - e.g., SAM (Cornia et al., 2018) performed very well on the first two fixations in comparison to the overall benchmark, which was best predicted by current, leading deep learning models such as SALICON, SalGAN, and DeepGazellE (Jiang et al., 2015; Linardos et al., 2021; Pan et al., 2017). This variability showed in different model rank orders between the first few fixations, relatively later fixations, or all 18 fixations, respectively (corresponding to the leftmost and the rightmost rank order in Figure 3 top right). Of course, a map of cumulative fixations contains more information whereas a map for only the first fixation is arguably more sparse - despite already 2,607 fixations contributing to it. Are differences in performance across fixations therefore driven by data sparsity? Fixation maps for only the first and the ninth fixation, respectively, (Figure 3 bottom right) compared with the overall map of all fixations (Figure 1 right), showed markedly different patterns as function of early- versus late viewing behavior: Fixation maps were highly focal for the first and much more spread out for later fixations, data sparsity cannot have driven these effects. The benchmark of models per fixation (Figure 3 bottom left) further showed relatively worse performance for early fixations. Around six to ten fixations marked the point of best performance, which roughly matches the overall number of fixation clusters observed. NSS scaled between centre prior and fixation map then dropped again across models. This challenges the current approach of optimizing saliency models on just one fixation map per image - in fact, what attracts gaze early on might be substantially different from what attracts gaze after a few fixations or extended viewing.”

For instance: How sensitive is maximum performance to the choice of smoothing kernel size?

We retested the Fixation Map with a doubled kernel size (ca. 2 dva; see heatmap below) and found that the match between the smoothed map and the discrete map was slightly lower, NSS = 0.66.

Because all other NSS scores were computed between each saliency map and the discrete fixation map, those absolute values did not change – and all scores are thus relatively higher compared to the maximum score of the Fixation Map, but otherwise unchanged.

The same holds for across-fixation analyses; only the maximum achievable score of the Fixation Map was slightly lower, but otherwise followed the same pattern as with a smaller smoothing kernel. All other absolute NSS scores were unchanged, but relatively scored slightly better compared to the maximum score.

See the following figure for reference (not included in the manuscript):

We now write in the newly added Limitations section:

“Whilst we did not find qualitative differences in results for two different kernel sizes to create the fixation maps, more extreme choices for kernels or flexible centre priors might affect the here reported bounds and thus findings.”

How dependent is normalised model performance on cut-off values when thresholding saliency maps? Could saliency models using the same set of features and weights improve normalised performance for the first fixations if they used more greedy thresholding on their saliency maps?

This is an interesting point. Whilst the NSS used here does not make use of thresholds, unlike some other performance metrics, saliency maps itself can be thresholded differently to give more/less weight to the most relevant image regions. Given the spatial distribution maps and more focal fixation patterns for the first fixations, a greedy thresholding approach for earlier vs later fixations might indeed solve this issue – by making a very specific prediction for the first fixations (with a greedy threshold) and a more spread-out prediction for later fixations (with ‘liberal’ thresholds). We thank the reviewer for this excellent idea that we have now added to the discussion as follows:

“One possibility to account for the variability in prediction quality across fixations lies in adjusting how saliency models apply thresholds: For first fixations, only the few most focal locations could be emphasized by greedier thresholds to represent the focal distribution of fixation locations observed early on. For subsequent fixations, a more liberal approach might be employed, allowing more spread-out predictions - as observed for intermediate or later fixations.”

How much of the higher entropy for later fixation maps is due to replicable inter-individual variance and how much to higher levels of randomness (i.e. in a repeat experiment would fixation 18 of subject A have a lower likelihood to be the same as fixation 1)?

This is a valid question. One can only speculate here given that there are no direct ways to test this. As described in the manuscript, the first fixation is quite focal. Given some degree of inhibition of return, there are only so many new locations that can be fixated, and so on, until our cut-off point of 18 fixations. As such, it is to be expected that the distribution of fixations will eventually be relatively homogeneous.

As to whether this follows a predictable pattern, it seems qualitatively that the first fixation is generally made on the paddleboarder or the kite surfer with low inter-individual variance (see images below), but already from the second fixation onward (non-cumulative) this becomes less focal and thus shows higher degrees of variance.

1st fixation

2nd fixation

3rd fixation

It would also be good to show model performance for individual fixations rather than cumulated across the first n fixations only (this is mentioned in the text for a comparison of fixation 9 and the first two, but I can't seem to find it in the figures).

We changed Figure 3 accordingly by replacing the lower left subplot and updated the text and caption accordingly.

This pattern is roughly reflected also in the absolute NSS values for individual fixations (not included in the manuscript, given below).

“The benchmark of models per fixation (Figure 3 bottom left) further showed relatively worse performance for early fixations. Around six to ten fixations marked the point of best performance, which roughly matches the overall number of fixation clusters observed. NSS scaled between central bias and fixation map then dropped again across models.”

We further adjusted the discussion and abstract correspondingly.

Discussion:

“Very late fixations, in turn, could disperse even further and be captured worse as a consequence. It is possible that the predictions for the first few fixations are less accurate,

because fixation allocation changes after approximately the first second, resulting in many intermediate and late fixations, but only relatively few early fixations in training data.”

Finally, some of the updated text appears to contain mistakes (e.g. legend of Figure 2 ‘scaled NSS scaled between...’), so a careful proofread would be in order. Figure S6 and the following page seem corrupted in the PDF

We fixed the typo and proofread the manuscript multiple times to catch any other mistakes.

12th Sep 23

Dear Dr Strauch,

Your manuscript titled "None to rule them all? Systematic shortcomings in modeling saliency across individuals and fixations" has now been seen again by Reviewer #2, whose comments appear below. In light of their advice I am delighted to say that we are happy, in principle, to publish a suitably revised version in Communications Psychology under the open access CC BY license (Creative Commons Attribution v4.0 International License).

We therefore invite you to revise your paper one last time to address the remaining concerns of our reviewers and a list of editorial requests. At the same time we ask that you edit your manuscript to comply with our format requirements and to maximise the accessibility and therefore the impact of your work.

EDITORIAL REQUESTS:

SUBMISSION INFORMATION:

OPEN ACCESS:

Communications Psychology is a fully open access journal. Articles are made freely accessible on publication under a [CC BY license](http://creativecommons.org/licenses/by/4.0) (Creative Commons Attribution 4.0 International License). This license allows maximum dissemination and re-use of open access materials and is preferred by many research funding bodies.

For further information about article processing charges, open access funding, and advice and support from Nature Research, please visit <https://www.nature.com/commspsychol/article-processing-charges>

At acceptance, you will be provided with instructions for completing this CC BY license on behalf of all authors. This grants us the necessary permissions to publish your paper. Additionally, you will be asked to declare that all required third party permissions have been obtained, and to provide billing

information in order to pay the article-processing charge (APC).

* TRANSPARENT PEER REVIEW: Communications Psychology uses a transparent peer review system. On author request, confidential information and data can be removed from the published reviewer reports and rebuttal letters prior to publication. If you are concerned about the release of confidential data, please let us know specifically what information you would like to have removed. Please note that we cannot incorporate redactions for any other reasons.

* CODE AVAILABILITY: All Communications Psychology manuscripts must include a section titled "Code Availability" at the end of the methods section. We require that the custom analysis code supporting your conclusions is made available in a publicly accessible repository at this stage; please choose a repository that generates a digital object identifier (DOI) for the code; the link to the repository and the DOI must be included in the Code Availability statement. Publication as Supplementary Information will not suffice.

* DATA AVAILABILITY:

[link redacted]

Best regards,

Marike Schiffer

Marike Schiffer, PhD
Chief Editor
Communications Psychology

on behalf of Eva Pool.

REVIEWERS' COMMENTS:

Reviewer #2 (Remarks to the Author):

I thank the authors for the more nuanced discussion of model performance and congratulate them on this great paper!

One final comment: I may miss something here, but find the following new sentence on p 12 hard to understand:

'It is possible that the predictions for the first few fixations are less accurate, because fixation allocation changes after approximately the first second, resulting in many intermediate and late fixations, but only relatively few early fixations in training data'

Is this referring to more *cumulated* fixations resulting in a larger training set? Or does the changing allocation of attention refer to the fact that the entropy in a group map increases after the first few fixations, leading to a higher number of peaks? Or something else entirely? Rephrasing may help to get the intended message across with greater clarity.

Response letter, our replies in green, edits in text in blue

REVIEWERS' COMMENTS:

Reviewer #2 (Remarks to the Author):

I thank the authors for the more nuanced discussion of model performance and congratulate them on this great paper!

We thank the reviewer for the thoughtful and constructive comments throughout the different versions of the manuscript.

One final comment: I may miss something here, but find the following new sentence on p 12 hard to understand:

'It is possible that the predictions for the first few fixations are less accurate, because fixation allocation changes after approximately the first second, resulting in many intermediate and late fixations, but only relatively few early fixations in training data'

Is this referring to more *cumulated* fixations resulting in a larger training set? Or does the changing allocation of attention refer to the fact that the entropy in a group map increases after the first few fixations, leading to a higher number of peaks? Or something else entirely? Rephrasing may help to get the intended message across with greater clarity.

We agree that this will benefit from clarification and now write:

Saliency models are optimized using cumulative fixation maps obtained from several seconds of free viewing. This practice might have introduced bias, as later fixations are disproportionately weighted in these maps relative to the initial two or three fixations.